# The Na,K-ATPase acts upstream of phosphoinositide PI(4,5)P₂ facilitating unconventional secretion of Fibroblast Growth Factor 2

Cyril Legrand [1,6], Roberto Saleppico [1,6], Jana Sticht[2,3], Fabio Lolicato[1,4,5], Hans-Michael Müller [1], Sabine Wegehingel[1], Eleni Dimou [1], Julia P. Steringer[1], Helge Ewers [2], Ilpo Vattulainen [4,5], Christian Freund[2] & Walter Nickel [1✉]

FGF2 is a tumor cell survival factor that is exported from cells by an ER/Golgi-independent secretory pathway. This unconventional mechanism of protein secretion is based on direct translocation of FGF2 across the plasma membrane. The Na,K-ATPase has previously been shown to play a role in this process, however, the underlying mechanism has remained elusive. Here, we define structural elements that are critical for a direct physical interaction between FGF2 and the α1 subunit of the Na,K-ATPase. In intact cells, corresponding FGF2 mutant forms were impaired regarding both recruitment at the inner plasma membrane leaflet and secretion. Ouabain, a drug that inhibits both the Na,K-ATPase and FGF2 secretion, was found to impair the interaction of FGF2 with the Na,K-ATPase in cells. Our findings reveal the Na,K-ATPase as the initial recruitment factor for FGF2 at the inner plasma membrane leaflet being required for efficient membrane translocation of FGF2 to cell surfaces.

[1] Heidelberg University Biochemistry Center, Im Neuenheimer Feld 328, 69120 Heidelberg, Germany. [2] Institute for Chemistry and Biochemistry, Freie Universität Berlin, Thielallee 63, 14195 Berlin, Germany. [3] Core Facility BioSupraMol, Freie Universität Berlin, Berlin, Germany. [4] Department of Physics, University of Helsinki, FL-00014 Helsinki, Finland. [5] Computational Physics Laboratory, Tampere University, FI-33100 Tampere, Finland. [6]These authors contributed equally: Cyril Legrand, Roberto Saleppico. ✉email: walter.nickel@bzh.uni-heidelberg.de

I n eukaryotic cells, the ER/Golgi-dependent secretory pathway represents the major mechanism of protein transport into the extracellular space[1–4]. However, additional mechanisms of protein secretion exist that have collectively been termed 'unconventional protein secretion'[5,6]. A typical example is Fibroblast Growth Factor 2 (FGF2)[7], a cell survival factor involved in tumor-induced angiogenesis[8,9]. Following secretion from tumor cells and their cellular microenvironment, FGF2 exerts its biological functions through both autocrine and paracrine signaling mediated by the formation of ternary complexes with FGF high-affinity receptors and heparan sulfates on cell surfaces.

Despite the biological functions FGF2 exerts in the extracellular space, it lacks a signal peptide required for transport along the classical, ER/Golgi-dependent secretory pathway[5,6,10,11]. Based upon biochemical reconstitution experiments and bulk measurements of FGF2 secretion from cells, unconventional secretion of FGF2 was found to be mediated by direct translocation of FGF2 across the plasma membrane[12–14]. This process has recently been visualized in real time in living cells employing single-molecule TIRF microscopy[12]. Unconventional secretion of FGF2 depends on interactions with the α1 subunit of the Na,K-ATPase[15], Tec kinase[16,17], and the phosphoinositide PI(4,5)P$_2$[17–19] at the inner plasma membrane leaflet as well as heparan sulfate chains of proteoglycans at the outer leaflet[20,21]. Consistently, residues in FGF2 that mediate interactions with PI(4,5)P$_2$ [K127, R128, and K133 (refs. [18,22])] and heparan sulfates [K133 (refs. [20,22])] as well as the residue that is phosphorylated by Tec kinase [Y81 (refs. [16,17])] have been identified and shown to be critical for efficient secretion of FGF2. In addition, two cysteine residues (C77 and C95) of FGF2 play a critical role in PI(4,5)P$_2$-dependent formation of membrane-inserted FGF2 oligomers[23] that represent dynamic intermediates of FGF2 membrane translocation[5,7]. These findings are in line with the observation that FGF2 retains a fully folded state during membrane translocation to the cell surface[24,25]. Recently, key steps of FGF2 membrane translocation were reconstituted using giant unilamellar vesicles with all components being purified to homogeneity[22]. The findings summarized above led to a model in which membrane-inserted FGF2 oligomers serve as intermediates in FGF2 membrane translocation. These are assembled in a PI(4,5)P$_2$-dependent manner at the inner leaflet and are disassembled at the outer plasma membrane leaflet mediated by membrane proximal heparan sulfates[10,11]. This model explains directional transport of FGF2 from the cytoplasm into the extracellular space in an ER/Golgi-independent manner[5,7]. The general mechanism of this pathway of unconventional secretion is also relevant to other unconventionally secreted proteins[5,6]. For example, a role for PI(4,5)P$_2$ has been reported for unconventional secretion of Tau, HIV-Tat, and Interleukin 1β[26–31]. In addition, in case of Tau, sulfated glycosaminoglycans on cell surfaces have been shown to have a critical function in trans-cellular spreading that is mediated by unconventional secretion of Tau from donor cells[27,32].

While the core mechanism of FGF2 membrane translocation is understood in great detail, the role of additional factors in unconventional secretion of FGF2 remains unclear. In previous studies, a role for the catalytic α1-subunit of the Na,K-ATPase in unconventional secretion of FGF2 has been proposed. Initial evidence was based on the observation that ouabain, an inhibitor of the Na,K-ATPase, blocks FGF2 transport into the extracellular space[33–35]. These findings were corroborated by the observation that an α1 mutant to which ouabain cannot bind is capable of restoring FGF2 secretion in the presence of ouabain[36]. Finally, α1 was one of the strongest hits in a genome-wide RNAi screen for gene products whose down-regulation inhibits FGF2 secretion from cells[15]. Interestingly, α1 has also been suggested to be involved in unconventional secretion of HIV-Tat[37]. However, as opposed to the detailed insight that is available about the core process of FGF2 membrane translocation, the precise function of α1 subunit of the Na,K-ATPase in unconventional secretion of FGF2 is unknown[5,7,38,39].

In the current study, we define a sub-domain in the cytoplasmic part of α1 that forms the binding surface for FGF2. Employing a chemical crosslinking approach, we identified a physical 1:1 complex of FGF2 and this minimal α1 domain as the main crosslinking product. Using NMR spectroscopy, we identified two lysine residues (K54 and K60) on the molecular surface of FGF2 that are involved in binding to α1. Using both docking studies and molecular dynamics (MD) simulations, we validated the role of K54 and K60 in a thermodynamically relevant model system. In this way, we also identified residues in α1 that are potentially engaged in the protein–protein interaction interface with FGF2. For one of them (D560), we could establish a direct role as the interaction with FGF2 was impaired in a D560N variant form of α1. Similarly, FGF2 variants lacking K54/K60 were found to be largely incapable of binding to α1 as shown by two different kinds of biochemical protein–protein interaction assays. These observations were further analyzed in cell-based secretion assays, demonstrating a drop in FGF2 secretion efficiency when K54 and K60 were replaced by glutamates. Using a recently established single-molecule TIRF microscopy assay designed to quantify FGF2 in the vicinity of the plasma membrane in living cells[12], we found that K54/60E mutants of FGF2 failed to get efficiently recruited at the inner leaflet of the plasma membrane. By contrast, a FGF2 variant form lacking the ability to bind to PI(4,5)P$_2$ was found at the inner leaflet at amounts comparable to the wild-type protein. These results demonstrate that FGF2 binding to α1 precedes recruitment of FGF2 by PI(4,5)P$_2$. Furthermore, as opposed to in vitro experiments demonstrating efficient binding of FGF2 to PI(4,5)P$_2$ with a $K_D$ of about 1 μM[17–19,22], these findings suggest that, in the context of intact cells, α1 is required for efficient binding of FGF2 to PI(4,5)P$_2$. Finally, we provide conclusive evidence that ouabain inhibits FGF2 secretion by impairing the interaction of FGF2 with the Na, K-ATPase at the plasma membrane of cells. We propose that α1 is an auxiliary factor that acts upstream of PI(4,5)P$_2$ to increase the efficiency of unconventional secretion of FGF2 by accumulating FGF2 at the inner plasma membrane leaflet.

## Results

**A sub-domain of the α1 subunit that interacts with FGF2.** The starting point of this study was to define a minimal sub-domain that contains the protein interaction surface for FGF2. The set of GST-fusion proteins carrying different forms of the cytoplasmic domain of α1 is shown in Fig. 1a. In addition to this schematic representation, the corresponding α1 domains are illustrated in a 3D model of full-length α1 (Fig. 1b). They contain either the complete (GST-α1-CD1-3) or various sub-domains of the cytoplasmic domain of α1 as indicated. Based upon a structural analysis, we identified a globular domain of about 20 kDa in α1 that contains a highly acidic molecular surface as a potential recruitment site for FGF2. This sub-domain is part of the third loop of the cytoplasmic domain of α1 (α1-subCD3) that contains the nucleotide binding site of the α1 subunit of the Na,K-ATPase[34,40]. A biochemical pull-down approach was used to quantify the interaction of FGF2 with the various GST-α1 fusion proteins (Fig. 1c). The construct containing the complete cytoplasmic domain of α1 (α1-CD1-3[15]) was used as a positive control. These experiments revealed that FGF2 binds with similar efficiency to α1-CD1-3, α1-CD3 (containing only the third loop of the cytoplasmic domain of α1), and the small sub-domain of loop 3 (α1-subCD3). By contrast, FGF2 did bind only weakly to

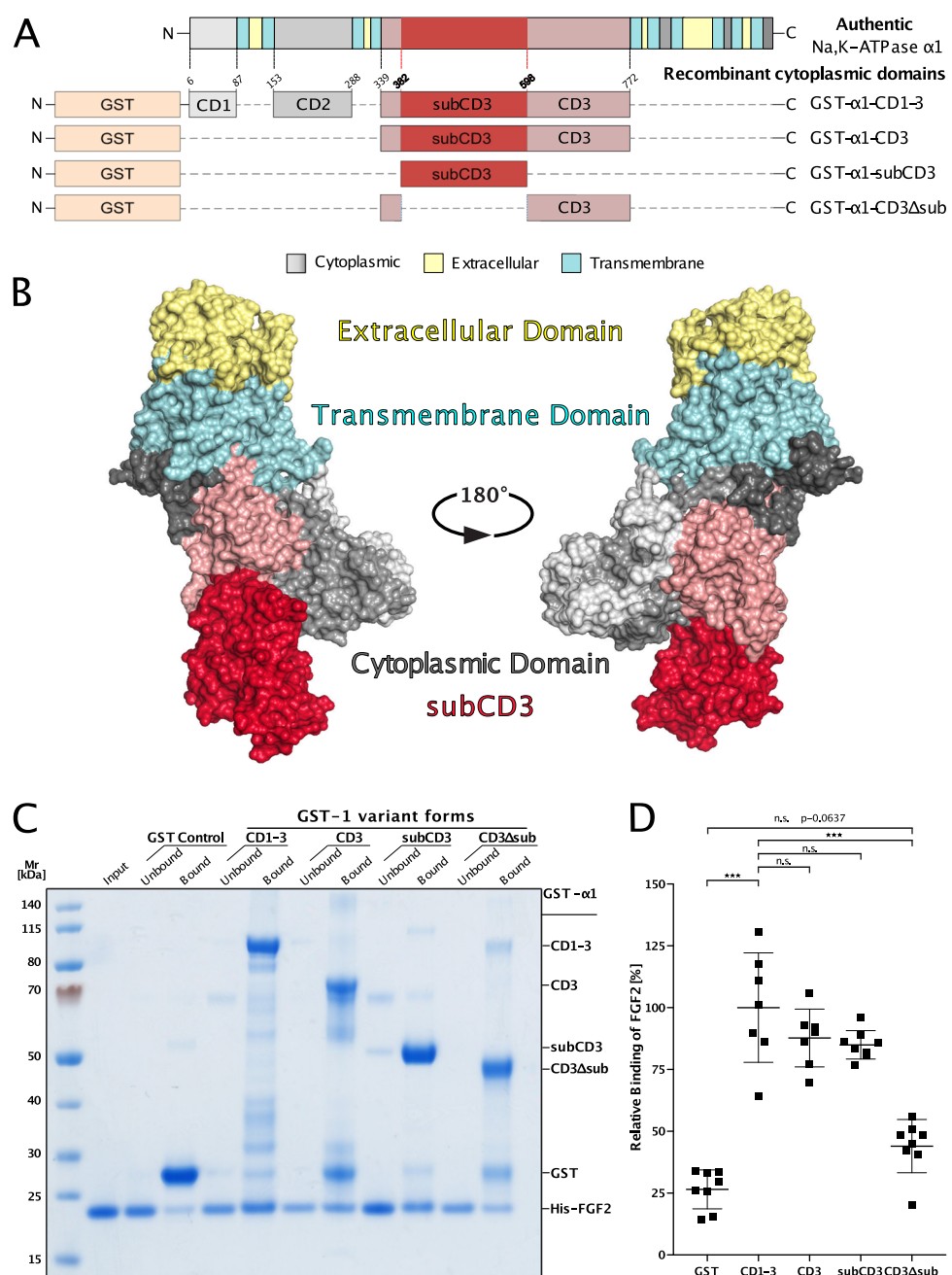

**Fig. 1 Identification of a sub-domain in the cytoplasmic domain of α1 that is both necessary and sufficient for binding of FGF2. a** Schematic representation of the different GST-tagged α1 constructs used in this study with a comparison to the complete α1 chain of the Na,K-ATPase. Short linkers were introduced between CD1 and CD2 as well as between CD2 and CD3 as described previously[15]. A short linker also exists in the α1-CD3Δsub construct connecting the N- and C-terminal parts of CD3. **b** Three-dimensional models of the complete α1 chain of the Na,K-ATPase based on the crystal structure of the Na,K-ATPase from *Sus scrofa*[47] (PDB ID: 3KDP). The different domains were annotated according to topology data from Uniprot (ID P05024, α1 structure from *Sus scrofa*). Transmembrane helices are shown in cyan, extracellular domains in yellow, and cytoplasmic CD1 and CD2 domains in shades of gray. The CD3 part of the cytoplasmic domain is colored in dark pink with the subCD3 region highlighted in red. **c** Biochemical pull-down experiments of FGF2 using GST-fusion proteins of various versions of the cytoplasmic domain of α1. Lane 1: FGF2 input (2.5%); lanes 2 and 3: GST control; lanes 4 and 5: GST-α1-CD1-3; lanes 6 and 7: GST-α1-CD3; lanes 8 and 9: GST-α1-subCD3; lanes 10 and 11: GST-α1-CD3Δsub. Bound (33% of each fraction) and unbound material (2.5% of each fraction) was analyzed by SDS-PAGE and Coomassie Blue protein staining. The results shown are representative for five independent experiments. **d** The intensity of the FGF2 protein bands from SDS-PAGE (panel **a**) was analyzed with the ImageStudio software package (LI-COR Biosciences). Ratios of bound versus total FGF2 were calculated and normalized to the amounts of FGF2 bound to GST-α1-CD1-3 containing the complete cytoplasmic domain of α1. Data are shown as mean ± SD ($n = 7$). ***$p \leq 0.001$.

both a construct lacking the sub-domain of loop 3 (GST-α1-CD3-ΔsubCD3) with GST alone used as a negative control (Fig. 1d).

The results from biochemical pull-down experiments were confirmed employing an AlphaScreen® protein–protein interaction

assay (Fig. 2). Cross-titration experiments using a wide range of combinations of concentrations of FGF2 and various α1 constructs demonstrated that FGF2 binds to the complete cytoplasmic domain (α1-CD1-3), loop 3 alone (α1-CD3) and α1-subCD3, the

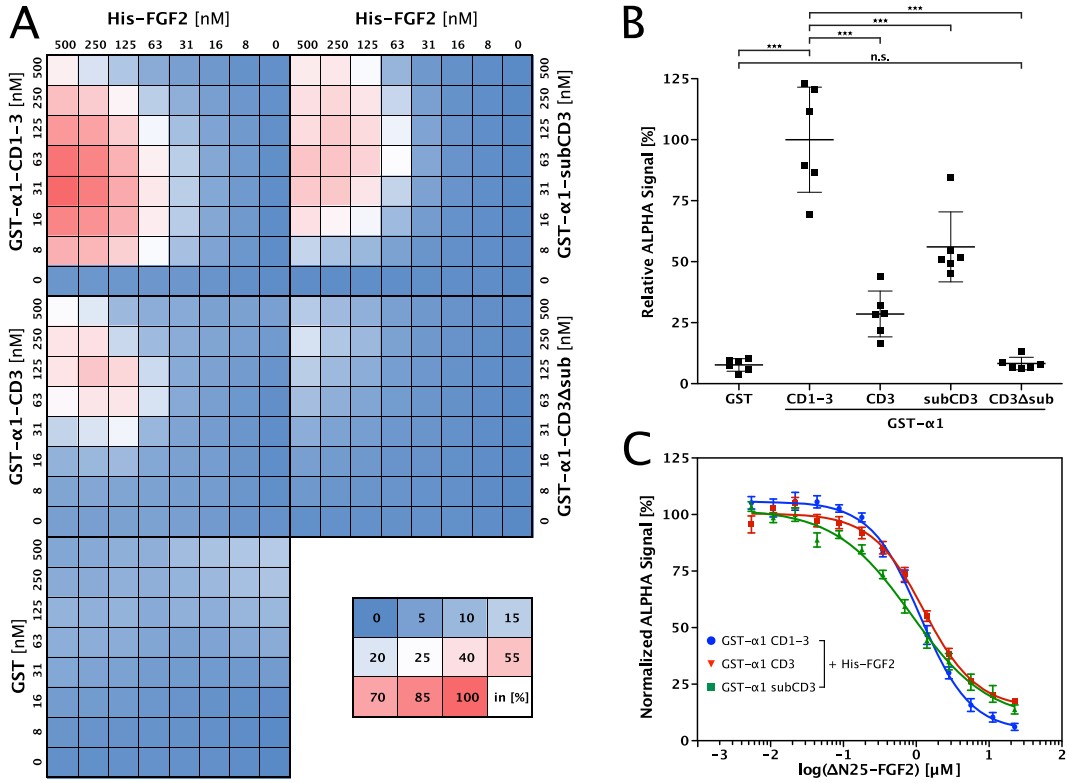

**Fig. 2 FGF2 binds to α1-subCD3 with sub-micromolar affinity as analyzed by an AlphaScreen protein–protein interaction assay. a** Cross-titration experiments conducted with His-tagged FGF2 and GST-tagged variant forms of the cytoplasmic domain of α1 as indicated. Data were normalized to the signal intensity measured for His-tagged FGF2-wt and GST-α1-CD1-3. The shown heat map is an average from six biological replicates. See Methods for details. **b** Quantification and statistical analysis of the relative alpha signal intensity from the cross-titration experiment with His-tagged FGF2 and the GST-fusion proteins of the cytoplasmic domain of α1 as indicated in panel **a**. Data are shown as mean ± SD ($n = 6$). ***$p \leq 0.001$. **c** Determination of affinity using $IC_{50}$ values from competition assays. GST-tagged α1 variant forms (CD1-3, CD3, or subCD3) and His-tagged FGF2 proteins were mixed and subjected to a serial dilution of ΔN25-FGF2 used as an untagged competitor. Data are shown as mean ± SEM ($n = 3$).

sub-domain of loop 3 of the cytoplasmic domain (Fig. 2a and quantified in Fig. 2b). By contrast, FGF2 binding to α1-CD3-ΔsubCD3 was similar to GST alone which was used as a negative control (Fig. 2b). Using an untagged version of FGF2 as a competitor for the interaction of His-tagged FGF2 and GST-tagged α1 constructs, the AlphaScreen® assay also allowed for a quantitative comparison of the affinities between FGF2 and α1-CD1-3, α1-CD3 as well as α1-subCD3, respectively (Fig. 2c[15,41]). As reported previously, FGF2 binds to the complete cytoplasmic domain of α1 with an apparent $K_D$ of about 0.8 μM[15]. With 1.1, 1.3, and 0.8 μM, similar apparent $K_D$ values were found for α1-CD1-3, α1-CD3 and α1-subCD3, respectively (Fig. 2c). These findings suggest that the sub-domain of loop 3 in the cytoplasmic domain of α1 contains the principal binding site for FGF2.

To determine the stoichiometry of the interaction between the minimal FGF2 binding domain in the cytoplasmic part of α1 (α1-subCD3) and FGF2, we conducted crosslinking experiments (Fig. 3). Two types of chemical crosslinkers were chosen targeting either cysteine (BMOE; Fig. 3a, b) or lysine (DSG; Fig. 3c, d) side chains. The formation of crosslinking products was monitored by both a western analysis using anti-FGF2 and anti-α1 antibodies (Fig. 3a, c) and an SDS-PAGE analysis based upon Coomassie protein staining (Fig. 3b, d). For both read-outs and crosslinkers, the main product was characterized by a migration behavior that is consistent with a molecular weight of about 46 kDa. This product was recognized by both anti-FGF2 and anti-α1 antibodies (Fig. 3a, c), suggesting a 1:1 complex of FGF2 and the α1-subCD3. For BMOE in particular, substantial amounts of the 1:1 crosslinking product were formed as it was readily detectable

by Coomassie protein staining as well (Fig. 3b). Additional crosslinking products with higher molecular weights were identified as well which are due to the ability of both FGF2 and α1 to form homo-oligomers (Fig. 3a–d; lanes 5 and 6). However, the formation of FGF2/α1 heterodimers occurred preferentially as they were formed at the expense of FGF2 homodimers (Fig. 3a–d, lane 7 versus lane 9). These results establish a direct physical contact between FGF2 and α1 with a heterodimer being the basic unit of this interaction.

**The binding interface between the Na,K-ATPase and FGF2.** In order to map the binding interface of α1-subCD3 on FGF2, NMR experiments were performed using an FGF2 variant form (FGF2-C77/95S) that is incapable of oligomerization[22,23]. $^1$H-$^{15}$N-HSQC spectra of $^{15}$N-labeled FGF2-C77/95S in the absence and in the presence of α1-subCD3 (in molar ratios of 1:1 and 1:2) were acquired and assignments were transferred from published data[42]. Chemical shift differences upon addition of α1-subCD3 were negligible for assigned peaks (<0.04 ppm, Supplementary Fig. 1) but some signals showed a clear reduction in peak intensity (Fig. 4, Supplementary Fig. 1). In agreement with the observed $K_D$ value of about 0.8 μM, this suggests a slow to intermediate exchange regime on the NMR timescale. The larger size of the complex compared to isolated FGF2 also manifests in a general line-broadening of signals that is seen as a small but clear reduction in peak intensities (Supplementary Fig. 1). Three of the six residues with strong reduction in signal intensities localize to the region spanning K54 to K60. The two effected lysine residues K54 and K60 are exposed on the molecular surface of FGF2 and,

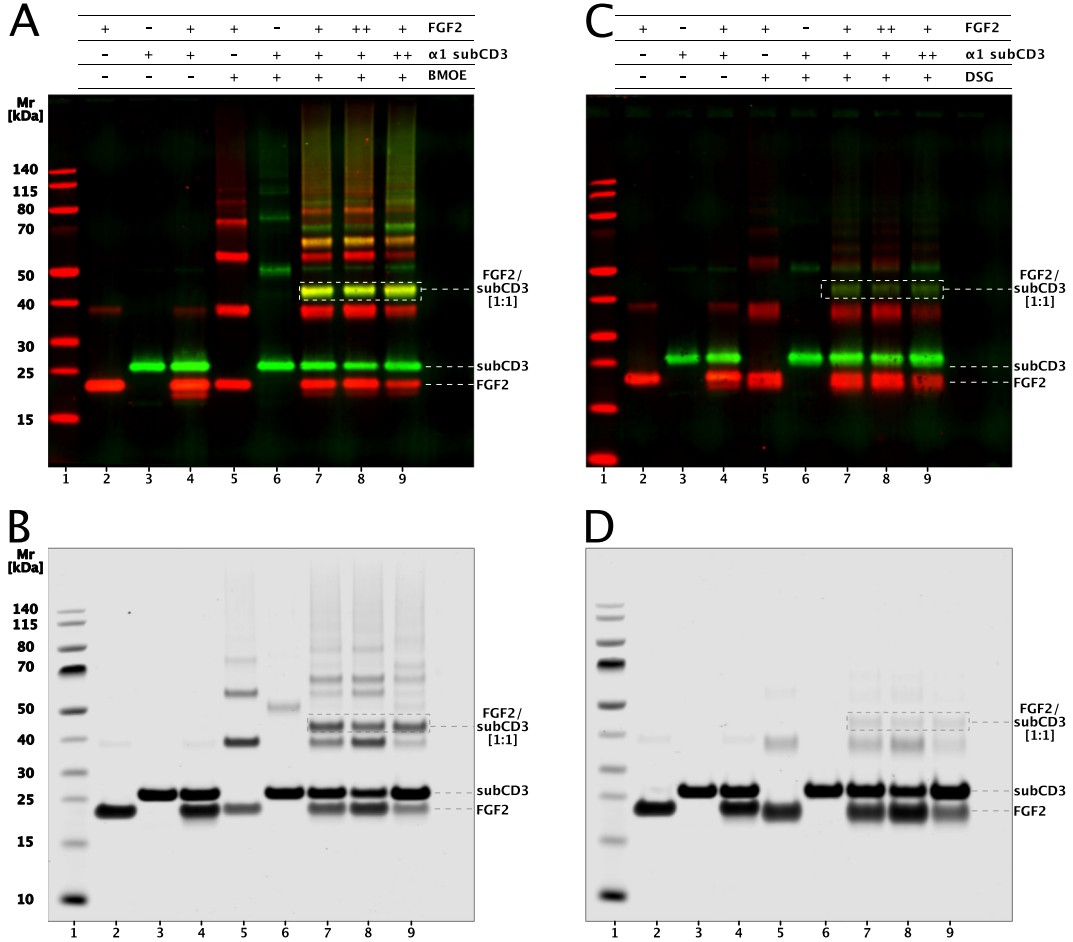

**Fig. 3 The basic binding unit between FGF2 and α1-subCD3 is a heterodimer as analyzed by chemical crosslinking experiments.** FGF2 and α1-subCD3 were mixed at a total concentration of 20 μM at molar ratios of 1:1 (+/+), 1:2 (+/++), and 2:1 (++/+) (FGF2/α1–subCD3). As chemical crosslinkers, bismaleidomethane (BMOE) and disuccinimidyl glutarate (DSG) were used at the molar ratios provided in Methods. Samples were analyzed by SDS-PAGE followed by either western blotting (**a**, **c**) or total protein staining using Coomassie. FGF2 and α1-subCD3 migrate with an apparent molecular weight of 18 and 27 kDa, respectively. The crosslinking product with an apparent stoichiometry of 1:1 shows a migration behavior that corresponds to a molecular weight of about 45 kDa as indicated. Lane 1: molecular weight standards; lane 2: input FGF2; lane 3: input α1-subCD3; lane 4: FGF2 plus α1-subCD3 without crosslinker; lane 5: FGF2 plus crosslinker; lane 6: α1-subCD3 plus crosslinker; lanes 7–9: FGF2 plus α1-subCD3 at 1:1, 2:1, and 1:2 stoichiometries in the presence of crosslinker. **a** BMOE: immunoblots using anti-FGF2 and anti-α1 primary antibodies. **c** DSG: immunoblots using anti-FGF2 and anti-α1 primary antibodies. **b** BMOE: total protein visualized by Coomassie staining. **d** DSG: total protein visualized by Coomassie staining.

therefore, were considered to potentially form a charged surface for the interaction with α1-subCD3.

In a previous study, two cysteine residues on the molecular surface of FGF2 (C77 and C95) were identified that are absent from all FGF family members with signal peptides[23]. These residues were demonstrated to play a critical role in unconventional secretion of FGF2 from cells[12,22,23]. As illustrated in Supplementary Fig. 2, similar to C77 and C95 (highlighted in yellow), the lysine residues in position 54 and 60 (highlighted in green) of FGF2 are absent from most FGF family members traveling through the ER/Golgi-dependent secretory pathway. By contrast, the majority of residues such as C33 and C100 are conserved throughout the FGF family irrespective of the mode of secretion of the respective FGF proteins, suggesting that they are important for example for the overall fold of FGF proteins. To test whether K54 and K60 in FGF2 are residues that are indeed critical for binding to the cytoplasmic domain of α1, we generated three FGF2 mutants in which these lysines were substituted by glutamate residues either separately or combined. In a first series of experiments, these FGF2 mutants were tested for binding to a GST-fusion protein containing the α1-subCD3 domain employing biochemical pull-down experiments

(Supplementary Fig. 3). A representative example is given in Supplementary Fig. 3a with the corresponding quantification and statistical analysis of four independent experiments shown in Supplementary Fig. 3b. While both the FGF2-K54E and the FGF2-K60E mutants already showed reduced binding efficiency, the interaction of FGF2-K54/60E to α1-subCD3 was strongly impaired.

These results were confirmed using the AlphaScreen® protein–protein interaction assay (Fig. 5). In panel a, a cross-titration experiment is shown analyzing a wide range of concentrations quantifying the interaction between α1-subCD3 and FGF2-wt, FGF2-K54E, FGF2-K60E as well as FGF2-K54/60E, respectively. Based on different combinations of concentrations resulting in the strongest interactions between FGF2-wt and α1-subCD3, a statistical analysis was conducted (Fig. 5b). Consistent with the biochemical pull-down experiments (Supplementary Fig. 3), single amino acid substitutions of either K54 or K60 caused partial inhibition of FGF2 binding to α1-subCD3. Substitution of both lysines by glutamates (FGF2-K54/60E) resulted in an almost complete lack of binding (Fig. 5b).

To test whether substitutions of K54 and K60 in FGF2 cause pleiotropic effects on other functional properties beyond binding

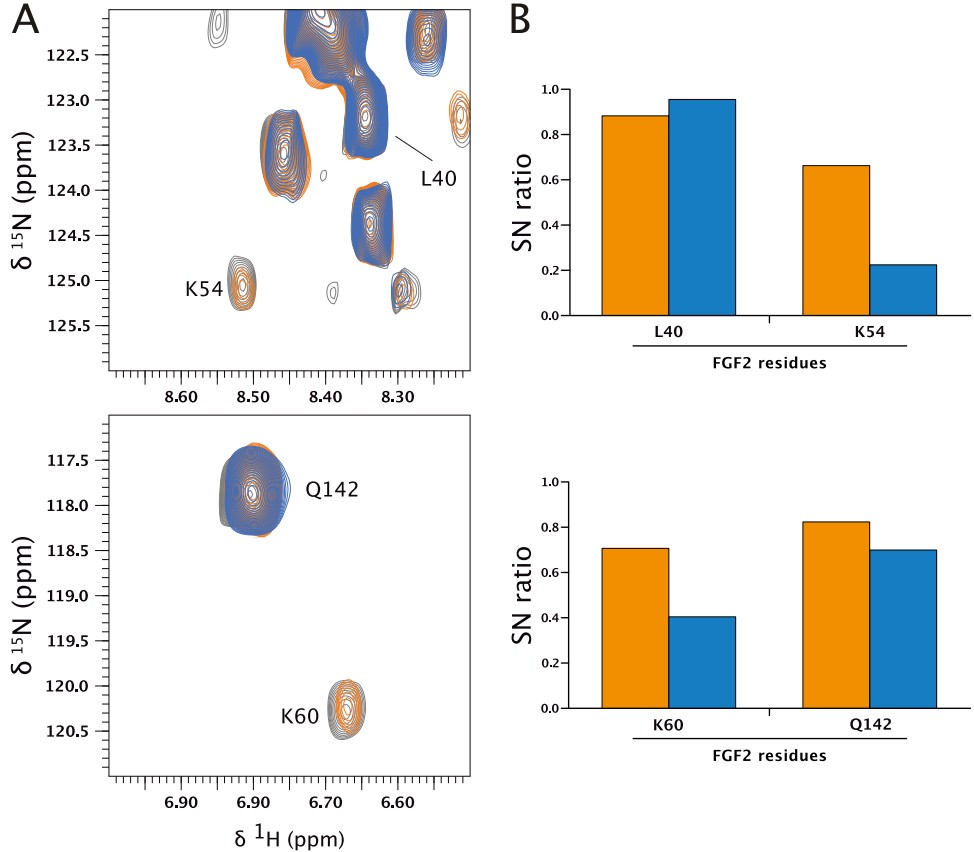

**Fig. 4 NMR analysis of the interaction between FGF2-C77/95S and α1-subCD3. a** Shown are two regions of the overlay of $^1$H-$^{15}$N-HSQC spectra of 77 μM $^{15}$N-labeled FGF2-C77/95S in the absence (gray) or presence of 70 μM (orange) or 138 μM α1-subCD3 (blue). **b** Signal-to-noise ratios (SN ratio) for indicated peaks of the spectra shown in **a** are plotted here with SN(+70 μM α1-subCD3)/SN(−α1-subCD3) in orange and SN(+138 μM α1-subCD3)/SN(−α1-subCD3) in blue.

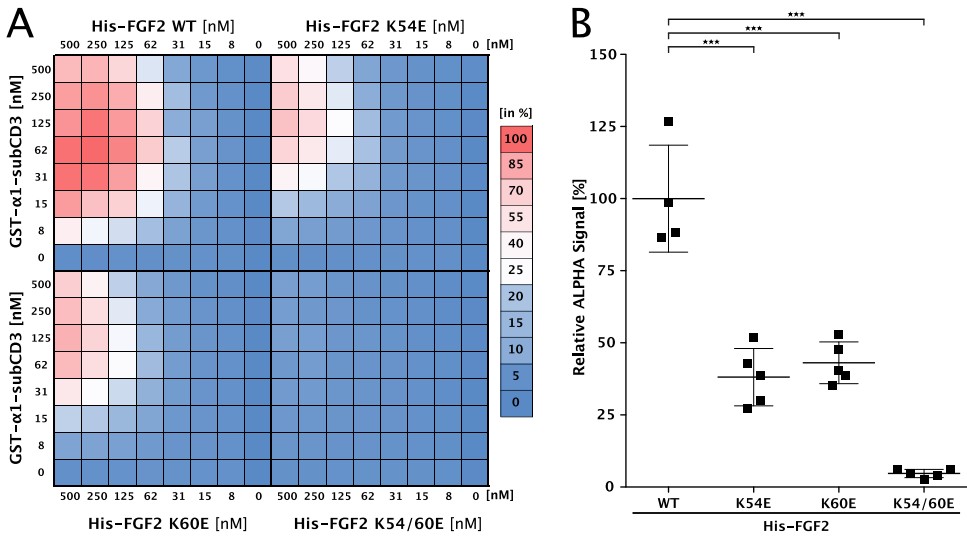

**Fig. 5 K54 and K60 are required for efficient binding of FGF2 to α1-subCD3 as analyzed by AlphaScreen protein–protein interaction experiments.**
**a** Cross-titration experiments conducted with various forms of His-tagged FGF2 and GST-tagged α1-subCD3. For each biological replicate, data were normalized to the signal measured for His-FGF2/GST-α1-CD1-3. As indicated by the color legend, data were represented as a heat map with the highest signal set to 100% (displayed in red) and the lowest signal set to 0% (displayed in blue). The shown heat map is an average from five biological replicates. For details, see Methods. **b** Quantification and statistical analysis of relative alpha signal intensities from the cross-titration experiment shown in panel **a**. Data are shown as mean ± SD ($n = 5$). ***$p \leq 0.001$.

to α1-subCD3, we compared all four variant forms of FGF2 (wt, K54E, K60E, and K54/60E) with regard to their ability to bind to PI(4,5)P$_2$ and to oligomerize (Supplementary Fig. 4a), to bind to heparin (Supplementary Fig. 4b) as well as tested for potential folding defects based upon thermal stability (Supplementary Fig. 4c). These experiments revealed that the substitution of K54 and K60 by glutamates does not have an impact on the parameters listed above.

**In silico analysis of the FGF2/α1–subCD3 binding interface**. To analyze the protein–protein interface between FGF2 and the cytoplasmic domain of Na,K-ATPase and to evaluate a potential role of K54 and K60 as residues in FGF2 that are in direct contact with α1-subunit, we conducted both in silico molecular docking studies and atomistic MD simulations (Fig. 6, Supplementary Movie 1). In a first stage, using protein–protein docking protocols, we scanned possible interaction interfaces between α1-subCD3 and the region in FGF2 exposing K54 and K60. Intensive molecular docking simulations were conducted by rotating FGF2 around the α1-subCD3 domain and calculating the corresponding interface scores, i.e. the differences in the energy state between the FGF2/α1–subCD3 complex versus unbound FGF2 and unbound α1-subCD3. The docking results were filtered and clustered based upon the root-mean-square deviation (RMSD) for FGF2 and the most representative structures of the largest populated clusters were refined employing MD simulations as explained in Methods. The ultimate goal of this approach was to find the most stable FGF2/α1–subCD3 interface and to characterize the network of interactions it involves. In Fig. 6a, the position of FGF2 relative to the full-length Na,K-ATPase is illustrated for the WT1 cluster. The human version of the α1-subCD3 domain was aligned to the same domain of the full-length crystal structure of the α1-subunit of the Na,K-ATPase from *Sus scrofa* (residues T380–V597; PDB ID: 3KDP). Membrane lipids [phosphatidylcholine and PI(4,5)P$_2$] were added to visualize the position of FGF2 bound to α1 relative to the inner leaflet of the plasma membrane. To test the stability of the interface between FGF2 and α1-subCD3, the central structure [which represents the structure with the smallest average value for the root-mean-square deviation (RMSD) compared to all other structures from the same cluster] of the cluster was simulated in three replicates for 200 ns. The probabilities of physical contacts as average contact maps of the FGF2/α1–subCD3 interface reveals a broad network of interactions stabilizing the interface of the WT1 system (Fig. 6b). As shown in the contact probability and interaction energy maps (Fig. 6d, f), the FGF2/α1–subCD3 interface is strongly stabilized by multiple electrostatic interactions between the residues E525–K60, E498–R30, K524–D27, D529-K74, and D560-K54. The contribution of each residue in stabilizing the FGF2/α1–subCD3 interface (Fig. 6b, c), calculated by summing up the contact probabilities for each residue individually, indicates that K54 and K60 have a high probability to be involved in the interaction.

The in silico results shown in Fig. 6 suggest the WT1 structure to be the most probable protein–protein interaction surface between FGF2 and α1-subCD3. It depends on K54 and K60 and is characterized by a large number of electrostatic interactions that are stable throughout the MD simulation time. Furthermore, MD simulations comparing FGF2-wt with FGF2-K54/60E provided direct evidence for the WT1 system being the relevant interface between FGF2 and α1-subCD3. This is evident from the observation that, in all MD simulations with FGF2-K54/60E (for details see Methods) and as opposed to FGF2-wt, FGF2-K54/60E did either dissociate from or was not able to form a stable interaction with α1-subCD3 (Supplementary Movie 1). In

additions, MD simulations of a variant form of α1-subCD3 (D560N) showed a reduction in the number of contact sites with FGF2 of about 70% (Fig. 6e). As shown in Fig. 6g, the D560N substitution is weakening the electrostatic network and, therefore, destabilizes the interaction interface between FGF2 and α1-subCD3. Thus, the molecular docking studies and MD simulations are consistent with the experimental data shown in Figs. 1–5.

To verify a potential role of D560 as well as other residues derived from the in silico studies described above, we generated the corresponding α1-subCD3 variant forms and tested them in protein–protein interaction assays based on AlphaScreen® technology (Fig. 7). Concentrations between 8 and 500 nM of both FGF2-wt and the α1-subCD3 variant forms indicated were analyzed in cross-titration experiments (Fig. 7a). As described in Methods, selected based on the strongest interactions between FGF2-wt and the wild-type form of α1-subCD3, various pairs of concentrations for both binding partners were used to quantify and statistically evaluate the interaction of all α1-subCD3 variant forms with FGF2-wt (Fig. 7b). While α1-subCD3 variants with substitutions of K524 and E525 did not show a phenotype under these conditions, the D560N substitution in α1-subCD3 resulted in a highly significant decrease of the interaction with FGF2 ($p$ value = 0.008) (Fig. 7b). All α1-subCD3 variant forms tested were characterized by normal thermal stability compared to the wild-type form of α1-subCD3 (Fig. 7c). These studies demonstrate D560 to represent, to our knowledge, the first structural element in the α1 subunit of the Na,K-ATPase that plays a direct role in the interaction with FGF2.

**A role for the Na,K-ATPase in FGF2 recruitment and secretion**. Using a single-particle TIRF microscopy approach established previously[12], we quantified FGF2-GFP recruitment at the inner plasma membrane leaflet of living cells. For the wild type and the various mutant forms of FGF2 indicated, both widefield and TIRF images were taken (Supplementary Fig. 5). While the widefield images allowed for the analysis of total expression levels of each of the FGF2-GFP fusion proteins indicated, the TIRF images were processed for the quantification of individual FGF2-GFP particles per surface area in the vicinity of the plasma membrane. In Supplementary Fig. 5a, FGF2-GFP fusion proteins are shown in which K54 and K60 were substituted for glutamates and compared with FGF2 wild-type. The results shown in Supplementary Fig. 5a along with those from real-time movies monitoring FGF2-GFP recruitment (Supplementary Movie 2) suggested that FGF2-K54/60E is impaired regarding physical contacts with the inner leaflet of the plasma membrane. A larger data set for each of the FGF2 mutants shown in Supplementary Fig. 5a and Supplementary Movie 2 was subjected to quantification and a statistical analysis as shown in Fig. 8. The number of FGF2-GFP particles in the vicinity of the plasma membrane were normalized to surface area with each data point representing the analysis of one cell. These experiments demonstrated that substitution of K54 and K60 by glutamates reduced the number of FGF2-GFP particles at the inner plasma membrane leaflet by about 50% (Fig. 8a, c). In additional experiments, the substitution of K54 and K60 was combined with mutations in the PI(4,5)P$_2$-binding site of FGF2 (Supplementary Fig. 5b, Fig. 8b, c, Supplementary Movie 3). These experiments revealed that mutations restricted to the PI(4,5)P$_2$-binding site in FGF2 alone did not affect FGF2-GFP recruitment at the inner plasma membrane leaflet (Fig. 8b, c). In addition, the combination of substitutions of K54/K60 and mutations in the PI(4,5)P$_2$-binding site of FGF2 caused a similar phenotype compared to what was observed for K54/60E alone (Fig. 8b, c). The differences in the efficiency of FGF2 recruitment at the inner leaflet between FGF2 variant forms

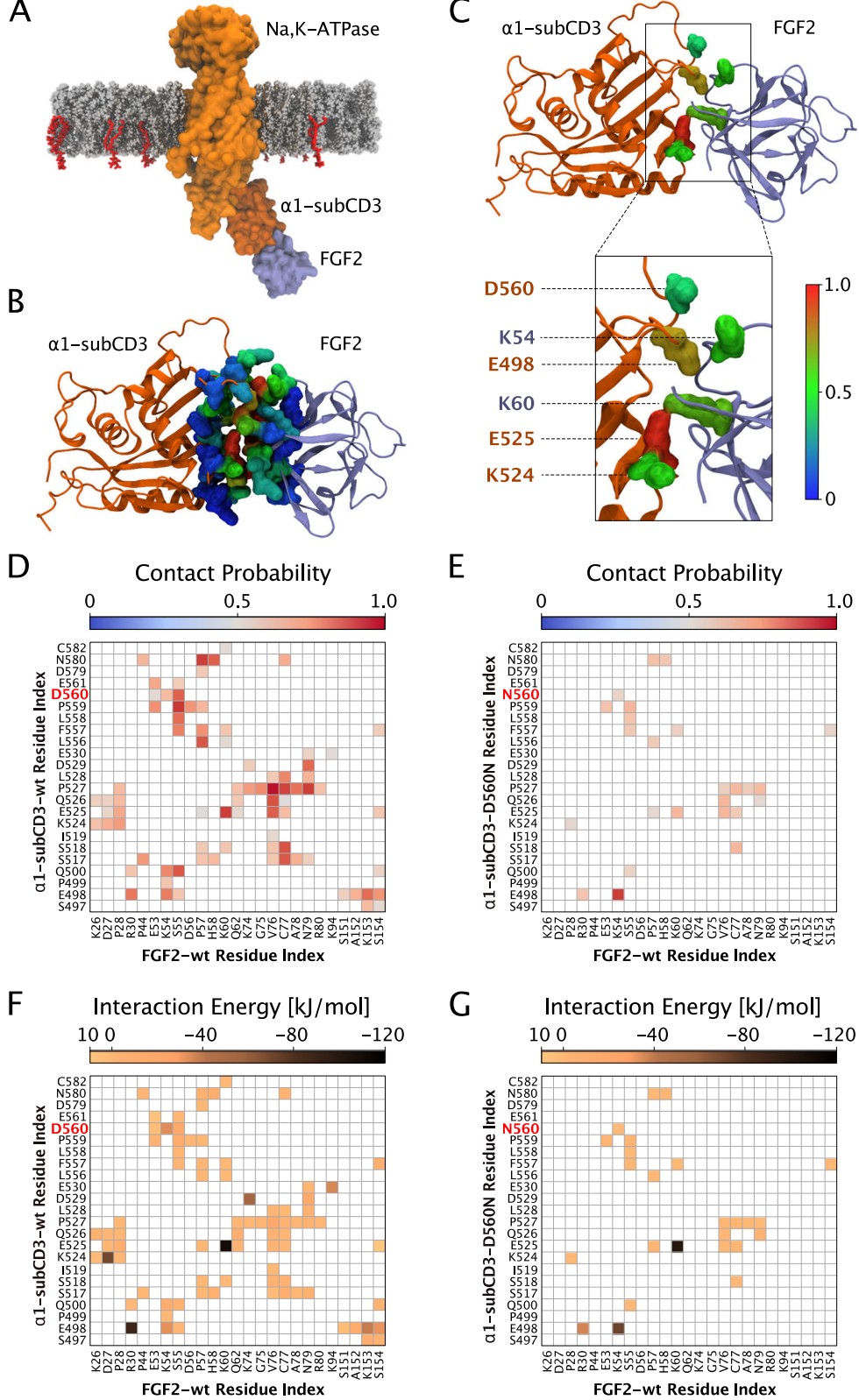

defective in either α1 binding (FGF2-K54/60E) or PI(4,5)P$_2$ binding (FGF2-K127Q/R128Q/K133Q) are particularly evident following GFP background correction (Fig. 8c). While substitution of K54 and K60 by glutamates reduced FGF2-GFP recruitment by almost 50%, the FGF2 mutant incapable of binding to PI(4,5)P$_2$ was fully functional with regard to recruitment at the inner plasma membrane leaflet. In addition, when both sets of

amino acid substitutions were combined, the observed phenotype was not stronger compared to the K54/60E substitution alone (Fig. 8c). These findings suggest that, during recruitment at the inner plasma membrane leaflet, the physical contact of FGF2 with α1 subunit of the Na,K-ATPase precedes the interaction of FGF2 with PI(4,5)P$_2$. Furthermore, in a cellular context, the data shown in Supplementary Fig. 5 and in Fig. 8 imply that the interaction of

**Fig. 6 Characterization of the molecular interface between FGF2 and α1-subCD3 based on in silico docking studies and atomistic molecular dynamics simulations. a** Representative structure of the WT1 cluster for in silico docking of FGF2 and α1-subCD3. **b** Average structure of the FGF2/α1-subCD3 interface. **c** Critical residues in the FGF2/α1-subCD3 interface including K54 and K60. **d** Pairwise contact map with all residues for the FGF2/α1-subCD3-wt interface. **e** Pairwise contact map with all residues for the FGF2/α1-subCD3-D560N interface. **f** Pairwise average interaction energy map for the FGF2/α1-subCD3-wt interface. **g** Pairwise average interaction energy map for the FGF2/α1-subCD3-D560N interface. In panel **a**, the most representative structure of the WT1 cluster is illustrated. The human α1-subCD3 domain was aligned to residues T380–V597 of the crystal structure of the α1-subunit of the Na,K-ATPase from Sus scrofa (PDB ID: 3KDP). The Na,K-ATPase is represented as an orange surface with the α1-subCD3 domain highlighted using a darker shade. FGF2 is shown as a violet surface. The PI(4,5)P$_2$ and phosphatidylcholine membrane lipids are represented using van der Waals spheres with red and gray colors, respectively. In panel **b**, an average structure of the FGF2/α1–subCD3 interface is shown illustrating the contribution to the interaction for each residue individually. It is defined as the sum of the probabilities of contacts for each residue and it is represented as a colored surface using the RGB color scale. Panel **c** highlights critical residues responsible for the FGF2/α1–subCD3 interaction including the FGF2 residues K54 and K60. Panels **d** and **e** show a pairwise contact map with all residues between FGF2/α1–subCD3-wt and FGF2/α1–subCD3-D560N, respectively. As a threshold, a probability of contact of more than 50% was set. Panels **f** and **g** show the average interaction energy (electrostatic and van der Waals contributions) of each residue pair between FGF2/α1–subCD3-wt and FGF2/α1–subCD3-D560N, respectively.

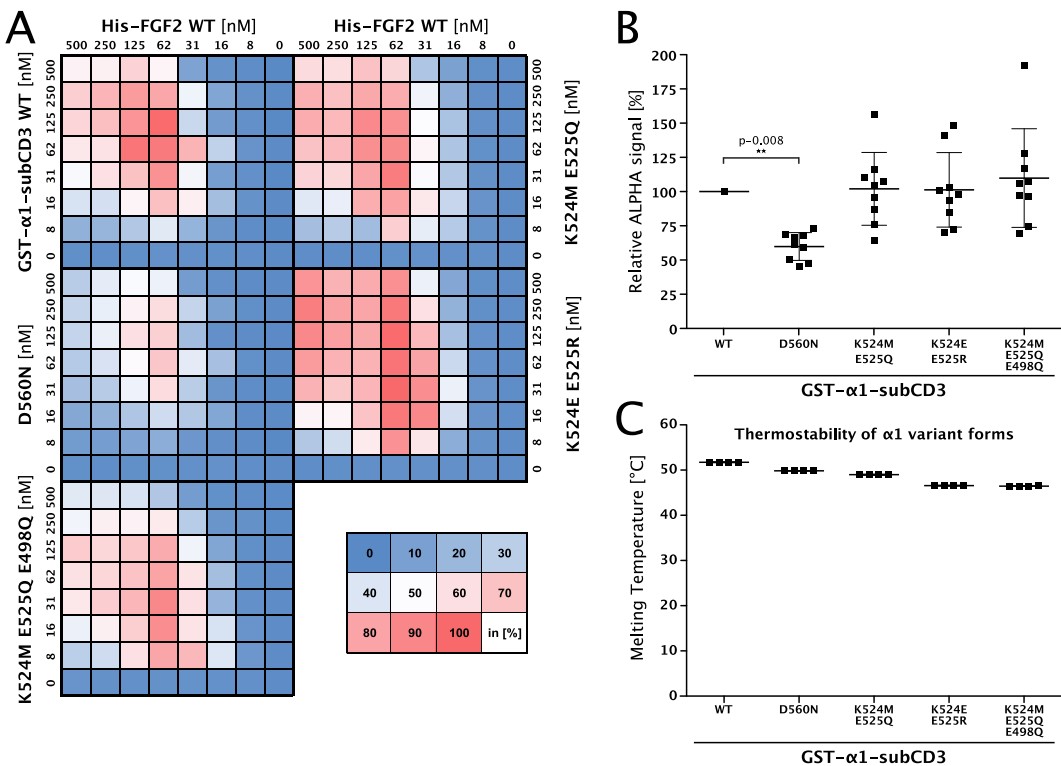

**Fig. 7 Structural elements in α1-subCD3 with relevance for the binding interface with FGF2. a** Cross-titration experiments conducted with His-tagged FGF2 and variant forms of GST-tagged α1-subCD3. A representative intensity map is shown with a color code for binding efficiency as indicated. **b** Quantification and statistical analysis of the relative alpha signal intensities comparing the interaction of His-tagged FGF2 with the variant forms of GST-tagged α1-subCD3 indicated. Data were normalized based on the signal intensity for FGF2/α1–subCD3. Standard deviations are shown. For details, see Methods. **c** Analysis of protein folding measuring thermal stability of α1-subCD3 variant forms. Protein samples of 10 μl at a final concentration of 3 mg/ml were analyzed by differential scanning fluorimetry (nanoDSF)[44]. Data are shown as mean ± SD ($n = 4$). For details, see Methods.

FGF2 with α1 facilitates the subsequent binding of FGF2 to PI(4,5)P$_2$, i.e. α1 acts upstream of PI(4,5)P$_2$.

We further tested whether amino acid substitutions that impair the interaction of FGF2 with α1 subunit of the Na,K-ATPase do affect secretion of FGF2-GFP to the cell surface (Fig. 9). These experiments included FGF2-GFP variant forms in which K54 and K60 were replaced by glutamates (Fig. 9a, b) and were also combined with amino acid substitutions (K127Q/R128Q/K133Q) that compromise the ability of FGF2 to bind to PI(4,5)P$_2$ (Fig. 9c, d). Following induction of protein expression with doxycycline, a cell surface biotinylation assay was used to quantify the extracellular amounts of the FGF2-GFP fusion proteins indicated[20,23,41,43]. These experiments revealed that a K54/60E substitution caused

a significant impairment of FGF2-GFP transport to the cell surface with only 60% efficiency compared to the wild-type form of FGF2-GFP ($p$ value ≤ 0.001) (Fig. 9a, b). When K54/60E and K127Q/R128Q/K133Q substitutions were combined, the efficiency of FGF2-GFP secretion dropped to less than 20% compared to the wild-type form of FGF2-GFP (Fig. 9c, d). This phenotype was similar to a mutant form of FGF2-GFP that lacks two cysteine residues (C77/95A) required for FGF2 oligomerization and the formation of membrane translocation intermediates[22,23]. Our combined findings suggest that efficient secretion of FGF2-GFP into the extracellular space is facilitated by a direct interaction of FGF2 with the cytoplasmic domain of α1 subunit of the Na,K-ATPase, a physical contact that precedes the interaction of FGF2 with PI(4,5)P$_2$.

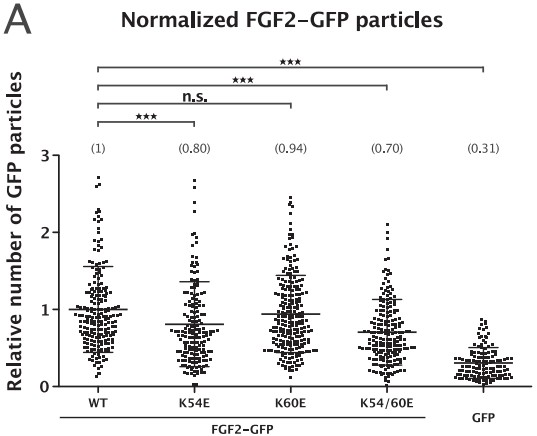

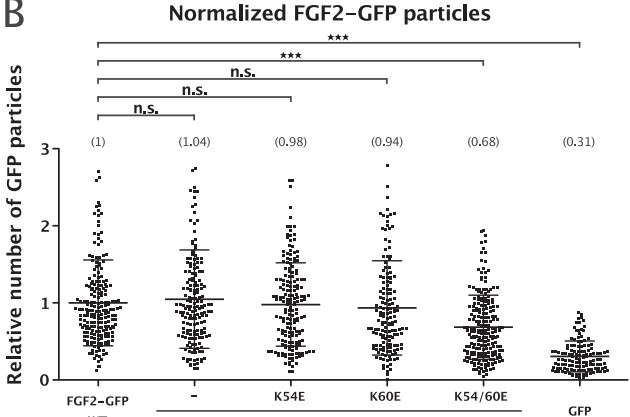

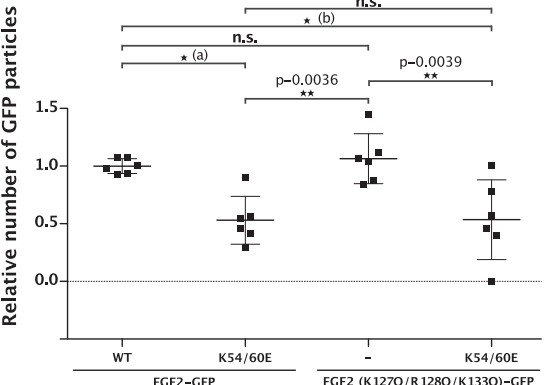

**Fig. 8 FGF2-GFP recruitment at the inner leaflet depends on direct interactions with the cytoplasmic domain of α1. a** FGF2-wt versus FGF2-K54/60E (FGF2 mutant deficient in binding to α1). **b** FGF2-wt versus FGF2-K54/60E in a K127Q/R128Q/K133Q background (FGF2 mutant deficient in binding to PI(4,5)P$_2$). **c** Direct comparison between FGF2-K54/60E and FGF2-K127Q/R128Q/K133Q following GFP background subtraction. Quantification of FGF2-GFP membrane recruitment at the inner leaflet of intact cells for all wild-type and mutant forms of FGF2 shown in panels **a** and **b** of Supplementary Fig. 5. Time-lapse TIRF movies with a total of 100 frames (100 ms/frame) were analyzed using the Fiji plugin TrackMate[12]. The number of GFP particles were normalized for both surface area and the relative expression levels of each FGF2 fusion protein in the corresponding cell line. In panel **a**, FGF2 mutants defective in binding to α1 are shown (K54E, K60E, and K54/60E). In panel **b**, the same mutants were combined with mutations in the PI(4,5)P$_2$ binding pocket of FGF2 (K127Q/R128Q/K133Q). The mean values of each condition are shown in brackets with the wild-type form of FGF2-GFP set to 1. Data are shown as mean ± SD ($n = 4$). ***$p \leq 0.001$. In panel **c**, the most important conditions were directly compared following GFP background subtraction and are shown as bar graphs. Data are shown as mean ± SD ($n = 6$). P value for (a) was 0.0106; p value for (b) was 0.0115.

inhibit the interaction between the cytoplasmic domain of α1 and FGF2 when analyzed with purified components in vitro[15]. The results shown in Fig. 10 demonstrate that ouabain binding to the membrane-spanning region of α1 inhibits recruitment of FGF2 which, in turn, results in efficient secretion of FGF2. These findings explain the ability of ouabain to impair FGF2 secretion from cells.

## Discussion

In the current study, we identified a sub-domain in the cytoplasmic part of α1 (α1-subCD3) that mediates a direct physical interaction with FGF2. The basic unit of this interaction is a heterodimer with a $K_D$ in the sub-micromolar range. The identification of α1-subCD3 as a minimal binding partner of FGF2 with about 25 kDa in size allowed for solution NMR experiments. This led to the identification of two lysine residues in position 54 and 60 on the molecular surface of FGF2 that were found critical for the interaction with α1-subCD3. Intriguingly, even though FGF family members in general are highly conserved reflecting the structural needs for building up the typical FGF fold, K54 and K60 represent FGF2-specific residues that are absent from most FGF family members carrying signal peptides for ER/Golgi-dependent protein secretion. Therefore, similar to what has previously been reported for two cysteine residues on the molecular surface of FGF2 (ref. [23]), the exclusive presence of both K54 and K60 in FGF2 points at a specific function of these residues in unconventional secretion of FGF2. Using various kinds of protein–protein interaction assays, we confirmed the NMR results demonstrating that FGF2 variant forms lacking K54/K60 are impaired in binding to α1-subCD3. These experiments were further validated by in silico docking studies and atomistic MD simulations demonstrating a role for K54/K60 in FGF2 binding to α1-subCD3 in a thermodynamically relevant model system. The use of molecular docking studies and MD simulations also allowed for identifying structural elements in the α1-subCD3 domain as components of the α1/FGF2 interface. With an aspartate residue (D560) in the subCD3 domain of α1, we could experimentally validate the, to our knowledge, first structural component known to play a direct role for the interaction between α1 and FGF2.

The biochemical and structural findings described above were found to be functionally relevant in a cell-based model system with FGF2 secretion being impaired in the absence of K54/K60. We made use of a recently established single-particle imaging

**Ouabain inhibits FGF2 recruitment by the Na,K-ATPase**. To elucidate the mechanism by which ouabain inhibits unconventional secretion of FGF2, we conducted experiments probing proximity of the α1 subunit of the Na,K-ATPase and FGF2 in a cellular context[15] (Fig. 10). As expected, the vast majority of proximity events between α1 and FGF2 (red dots in Fig. 10a–c) were found in the vicinity of the plasma membrane (Fig. 10a, b). Experiments with increasing concentrations of ouabain between 12 and 50 μM were conducted against a mock control. As shown in the statistical analysis depicted in Fig. 10d, ouabain caused a significant decrease of α1/FGF2 proximity events in a concentration-dependent manner ($p$ value $\leq 0.001$). Since ouabain binds to α1 within its membrane-spanning domain, it is not capable of being a direct competitor of FGF2 binding to the subCD3 domain in the cytoplasmic part of α1. Consistently, we previously demonstrated that ouabain does not

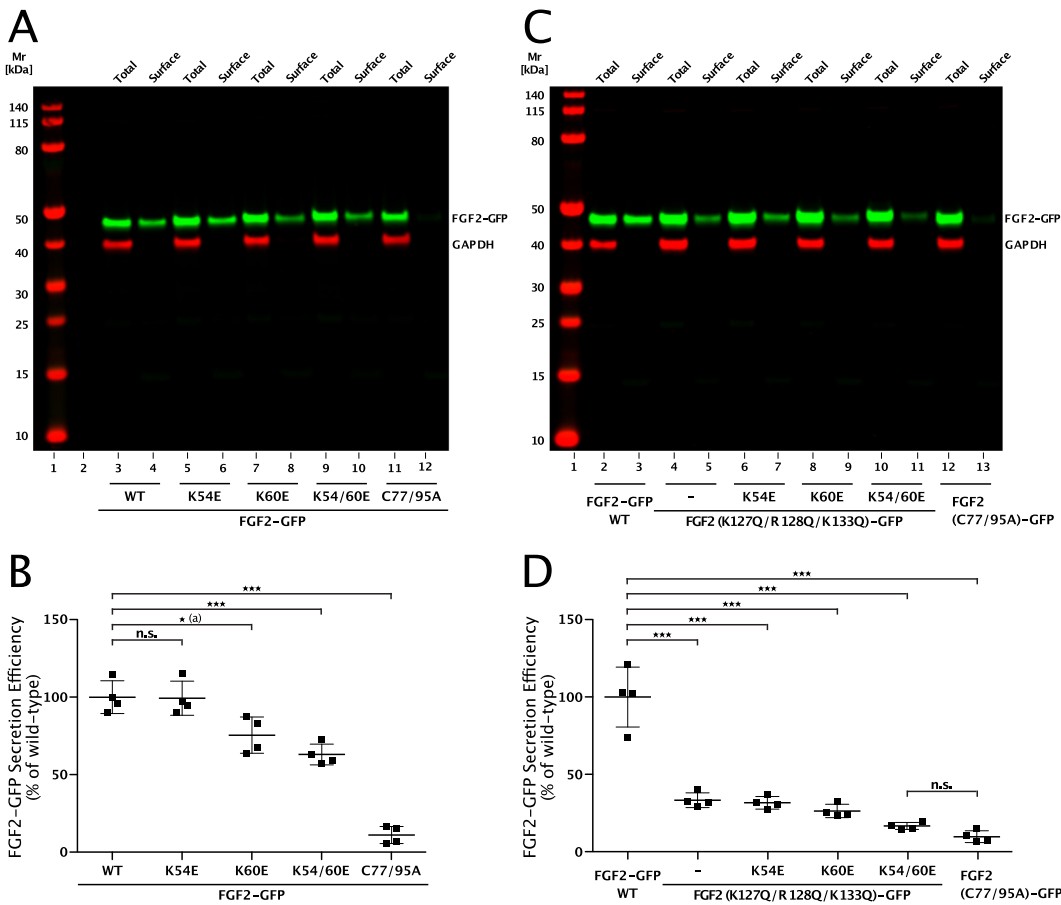

**Fig. 9 Efficient secretion of FGF2 from cells is facilitated by its interaction with α1-subunit of the Na,K-ATPase. a** Cell surface biotinylation experiments were conducted as described in Methods using stable CHO-K1 cell lines expressing either FGF2-wt-GFP, FGF2-K54E-GFP, FGF2-K60E-GFP, FGF2-K54/60E-GFP, or FGF2-C77/95A-GFP in a doxycycline-dependent manner. Aliquots from the total cell lysate (1.6%) and from the biotinylated fraction (33.3%; corresponding to the cell surface population of proteins) were subjected to SDS-PAGE and western blotting. Anti-GFP antibodies were used to detect the various FGF2-GFP fusion proteins indicated. Anti-GAPDH antibodies were used to detect intracellular GAPDH as a control for cell integrity during cell surface biotinylation. Primary antibodies were detected by fluorophore-labeled secondary antibodies and quantified using the Odyssey® CLx Imaging System (LI-COR Biosciences). **b** The efficiency of FGF2-GFP secretion of each variant form shown in panel **a** was quantified and normalized to the wild-type form that was set to 100%. Data are shown as mean ± SD ($n = 4$). $P$ value for a was 0.0161. ***$p \leq 0.001$. **c** Stable CHO-K1 cell lines expressing either FGF2-wt-GFP, FGF2-K127Q/R128Q/K133Q-GFP, FGF2-K54E/K127Q/R128Q/K133Q-GFP, FGF2-K60E/K127Q/R128Q/K133Q-GFP, FGF2-K54/60E/K127Q/R128Q/K133Q-GFP, or FGF2-C77/95A-GFP in a doxycycline-dependent manner were analyzed by cell surface biotinylation as described in the legend to panel **a**. **d** The efficiency of secretion of each variant form of FGF2-GFP shown in panel **c** was quantified and normalized as described in the legend to panel **b**. Data are shown as mean ± SD ($n = 4$). ***$p \leq 0.001$.

system studying FGF2 membrane recruitment at the inner plasma membrane leaflet employing real-time TIRF microscopy in living cells[12]. Intriguingly, we found that a FGF2 mutant lacking both K54 and K60 as well as the three critical residues (K127/R128/K133) of the PI(4,5)P$_2$-binding pocket was indeed impaired in binding to the inner leaflet. However, when the two sets of mutations were looked at separately, only the substitution of K54/K60 by glutamates caused a decrease in FGF2 recruitment at the inner leaflet. These findings have two important implications with (i) FGF2 binding to the α1-subunit of the Na,K-ATPase precedes FGF2 binding to PI(4,5)P$_2$ and (ii) the physical contact of FGF2 at the inner leaflet with α1 facilitates subsequent interactions of FGF2 with PI(4,5)P$_2$. Thus, beyond the identification of both a sub-domain in the cytoplasmic part of α1 and residues on the surface of FGF2 required for a direct interaction between these proteins, our findings suggest a function of α1 as an auxiliary factor in unconventional secretion of FGF2 that increases the efficiency of PI(4,5)P$_2$-dependent FGF2 oligomerization and membrane translocation. In this context, it is an interesting observation that, as opposed to in vitro experiments in which

FGF2 can bind to PI(4,5)P$_2$ in the absence of other factors[18,19], α1 appears to be important in intact cells to enable efficient binding of FGF2 to PI(4,5)P$_2$. Thus, the α1-subunit of the Na,K-ATPase may serve as a factor that accumulates FGF2 at the inner plasma membrane leaflet, facilitating PI(4,5)P$_2$-dependent FGF2 membrane translocation to the cell surface. Furthermore, it is an interesting hypothesis for future studies as to whether this function is linked to a modulation of the activity of the Na,K-ATPase to maintain a functional Na/K gradient across the plasma membrane during active events of FGF2 membrane translocation, a process that may transiently disturb the integrity of the plasma membrane. Finally, our study also sheds light on the mechanism by which ouabain inhibits FGF2 secretion. While ouabain does not affect the interaction of recombinant forms of the cytoplasmic domain of α1 and FGF2 in vitro[15], we found that, in a cellular context, ouabain inhibits α1/FGF2 proximity events at the plasma membrane. These findings suggest that, through ouabain binding to the transmembrane region of α1, conformational changes occur in the cytoplasmic domain that impair recruitment of FGF2. With this interaction being important for efficient

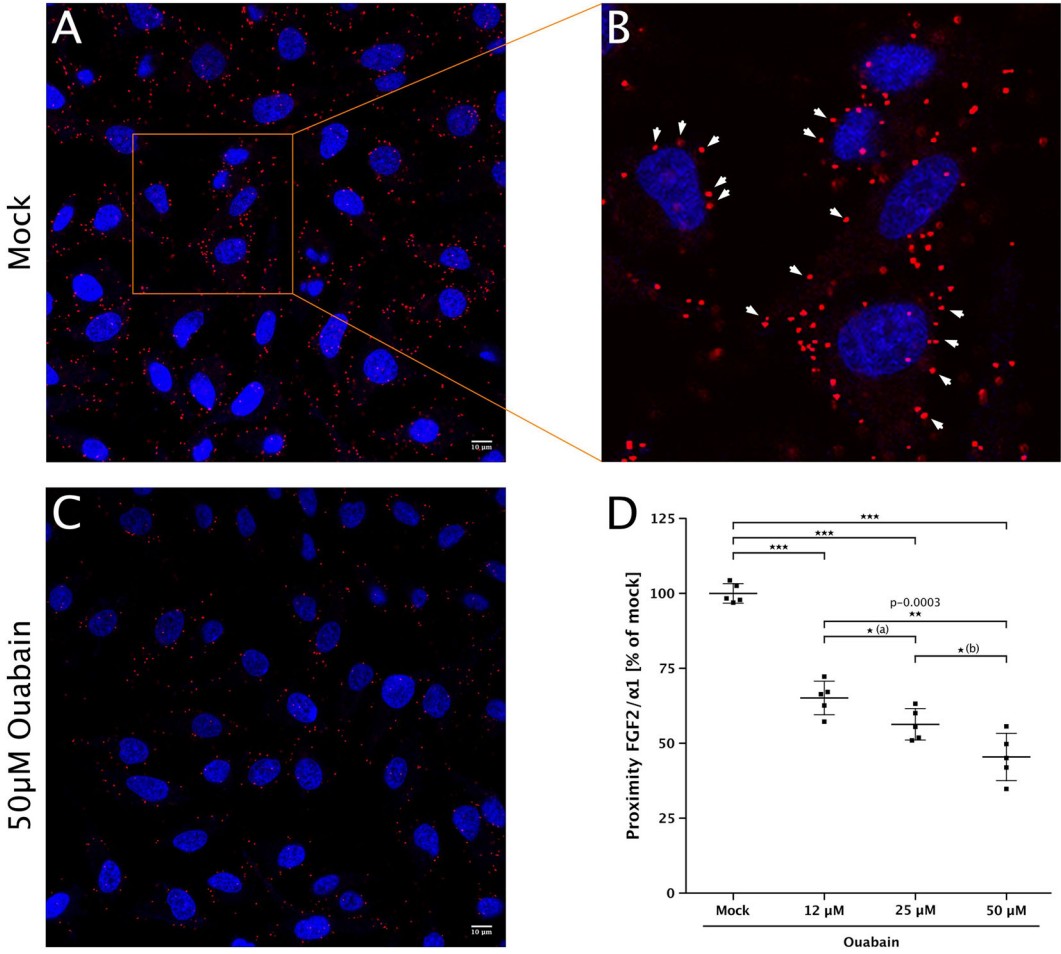

**Fig. 10 Ouabain inhibits proximity events between the α1-subunit of the Na,K-ATPase and FGF2 in a cellular context.** DuoLink assays (PLA®; Sigma-Aldrich) were conducted to quantify proximity of the α1 chain with FGF2 in a cellular context as described previously[15]. Cells were incubated with ouabain for 2 h at the concentrations indicated. Cells were fixed with acetone. Nuclei were labeled with Sytox green (Life Technologies) and cells were imaged by confocal microscopy. For further details, see Methods. **a** Representative example of mock-treated HeLa cells. Scale bar = 10 μm. **b** Selected area of panel **a** at a higher magnification. DuoLink proximity events (red dots) in the vicinity of the plasma membrane are labeled with white arrowheads. **c** Representative example for cells treated with 50 μM ouabain. **d** Quantification, normalization, and statistical analysis of α1/FGF2 proximity events in the absence and presence of ouabain at the concentrations indicated. Images were analyzed with the DuoLink Image Tool Software (Olink Bioscience). Background signals were subtracted and data were normalized relative to the mock control. Data are shown as mean ± SD ($n = 5$). P value for (a) was 0.1124; p value for (b) was 0.0381. ***$p \leq 0.001$.

secretion of FGF2 from cells, our results provide a plausible explanation of how ouabain inhibits this process.

Recently, beyond the established function of α1 in unconventional secretion of FGF2, evidence has been reported for a similar role of α1 in the secretion of HIV-Tat from HIV-infected T cells[37]. Along with the well-established role for PI(4,5)P$_2$ in unconventional secretion of HIV-Tat[30,31], these findings point at a shared secretory mechanism between FGF2 and HIV-Tat. Furthermore, with Tau and Interleukin 1β, two additional extracellular factors secreted by unconventional means, even more examples have been reported for unconventional secretory processes that depend on PI(4,5)P$_2$ as a prerequisite for their transport into the extracellular space[27–29,32]. Thus, after decades of research on unconventional secretory mechanisms in mammalian cells, common features between pathways that are taken by proteins functionally as different as FGF2, Interleukin 1β, Tau, and HIV-Tat are beginning to emerge.

## Methods
**Biochemical protein–protein interaction assays**. Recombinant proteins were expressed in *E. coli* and purified according to standard procedures. The 18 kDa

isoform of FGF2 was expressed and purified as a N-terminally His$_6$-tagged protein. For competition assays, a non-tagged form of FGF2 lacking 25 residues at the N-terminus (ΔN25-FGF2) was expressed and purified. For NMR spectroscopy, a monomeric FGF2 variant form (C77/95S) was expressed in M9 minimal medium with $^{15}$N-NH$_4$Cl as the sole nitrogen source in order to produce FGF2 as a $^{15}$N-labeled protein. In addition, recombinant forms of FGF2 were expressed and purified in which K54 and K60 were substituted by glutamates as indicated.

Four variant forms of the cytoplasmic domain of α1 were expressed as N-terminal GST-fusion proteins. This included a fusion of the three main loops of the cytoplasmic domain [GST-α1-CD1-3 (ref. [15])], the third loop of the cytoplasmic domain alone [GST-α1-CD3; T339-L772 from human α1][15], a small sub-domain of loop 3 (GST-α1-subCD3; T382–A598 from human α1), and the third cytoplasmic loop of α1 lacking the above-mentioned sub-domain of loop 3 (GST-α1-CD3Δsub; C343-L381 linked to A598-L772 from human α1). As shown in Figs. 6 and 7, based on MD simulations, several GST-tagged variant forms of α1-subCD3 were generated carrying amino acid substitutions in the potential α1/FGF2 interface.

To test the ability of FGF2 variant forms to bind to PI(4,5)P$_2$ concomitant with oligomerization, FGF2 (10 μM) was incubated with liposomes (2 mM total lipid) with a plasma-membrane-like lipid composition containing 3 mol% of PI(4,5)P$_2$. As described previously[17,18], experiments were conducted in a final volume of 50 μl HK buffer (150 mM KCl, 25 mM HEPES, pH 7.4) in an Eppendorf Thermomixer at 25 °C at 500 rpm. After 4 h of incubation liposomes were sedimented for 10 min at 16,000$g$, and washed with 100 μl HK buffer. Pellets were dissolved in 20 μl non-reducing SDS sample buffer and heated for 10 min at 65 °C. Samples were analyzed on 1.0 mm NuPAGE 4–12% non-reducing Bis-Tris gels/MES-running buffer

(Invitrogen) and proteins were stained with Coomassie InstantBlue (Expedeon). Input lanes contained 50% of FGF2 used in the oligomerization assays.

FGF2 variant forms were further compared to FGF2-wt with regard to their ability to bind to heparin. Heparin Sepharose 6 Fast Flow (GE Healthcare) beads (10 µl slurry) were washed four times in phosphate-buffered saline (PBS) and incubated for 1 h at room temperature with the FGF2 variant forms indicated (15 µg protein each in a total volume of 200 µl PBS). Following sedimentation of the beads and extensive washing, binding efficiency was analyzed by SDS-PAGE and protein staining using Coomassie InstantBlue (Expedeon).

To determine native protein folding by means of thermal stability, the various FGF2 and α1 variant forms used in this study were analyzed in nanoDSF experiments using a Prometheus NT 48 instrument (Nanotemper)[44]. This procedure monitors the light absorbance of proteins at 330 and 350 nm along a thermal gradient. The ratio of $A_{330nm}/A_{350nm}$ was plotted as a function of the temperature. This allows for determining the unfolding transition midpoint, i.e. the melting temperature of a protein. The FGF2 and α1 variant forms indicated were used in a volume of 10 µl at a final concentration of 1.5 mg/ml.

**Biochemical pull-down experiments.** For this set of experiments, Glutathione Sepharose beads were equilibrated in PBS buffer and incubated with the GST-tagged variants of α1 as indicated. GST alone was used as a negative control. GST-coupled Sepharose beads were blocked with 3% (w per v) BSA in PBS supplemented with 1 mM Benzamidine and 0.05% (w per v) Tween 20 (buffer A), washed extensively in buffer A. Finally, beads were resuspended with five bed volumes of buffer A. Per experimental condition, 75 µl of beads solution were incubated with 15 µg of His-tagged FGF2 in a total volume of 200 µl of buffer A for 1 h at room temperature. Following collection of beads by low-speed centrifugation and extensive washing with buffer A, bound protein was eluted with SDS sample buffer. Both bound (33%) and unbound (2.5%) material was analyzed by SDS-PAGE followed by protein staining using Coomassie InstantBlue. Gels were scanned using the Odyssey® CLx Imaging System (LI-COR Biosciences) and band intensities were quantified using the LI-COR ImageStudio software. For quantification, the ratios between bound and unbound material were calculated for each experimental condition as indicated.

**AlphaScreen-based protein–protein interaction assays.** For cross-titration of a wide range of concentrations of FGF2 and α1 variants and to determine affinity between FGF2 and α1 the AlphaScreen® protein–protein interaction assay was used [Figs. 2, 5 and 7 (ref. [15])]. N-terminally His-tagged FGF2 (wild type or mutants as indicated) and N-terminally GST-tagged α1 variants were cross-titrated from 500 to 8 nM in PBS supplemented with 0.1% (w per v) BSA and 0.05% (w per v) Tween 20. These experiments were conducted in 384-well plates. After 1 h of incubation, the proteins were incubated with AlphaScreen Ni-NTA acceptor and AlphaScreen® Glutathione donor beads, each at a final concentration of 11.25 µg/ml in a total volume of 15 µl. Following two additional hours of incubation, samples were measured using an EnVision plate reader (PerkinElmer).

For each biological replicate, data were normalized to the signal measured for His-FGF2/GST-α1 CD1-3 (Fig. 2) or for His-FGF2 WT/GST-α1-subCD3 (Figs. 5 and 7). As indicated in the supplemental raw data source file, a range of concentration pairs was used for the quantification and statistical analysis of the FGF2 and α1 variant forms indicated. Data were represented as a heat map with the highest signal set to 100% (displayed in red) and a value of 0 set to 0% (displayed in blue) as indicated by the color legend.

To determine binding affinity between His-FGF2-wt and GST-α1 variants as indicated, competition experiments were conducted. An untagged and N-terminally truncated form of FGF2 (ΔN25-FGF2) was used as a competitor which allowed for the determination of IC50 values. For these analyses, optimal concentrations of FGF2 (62 nM) and α1 (15 nM) were used based upon the cross-titration experiments described above. The competitor ΔN25-FGF2 was used in a concentration range between 22.5 µM and 1.4 nM. The respective pairs of His- and GST-tagged proteins were mixed with the ΔN25-FGF2 competitor in a final volume of 10 µl at the concentrations indicated. Addition of Alpha beads and measurements were done as described above. For each protein pair, the median signal of three technical replicates was calculated, normalized to the signal of the buffer lane, and plotted against the concentration of the ΔN25-FGF2 competitor. For each protein pair tested in competition experiments, the ALPHA signal was normalized to the median ALPHA signal of the buffer lane. Normalized data from three biological replicates were eventually plotted against the concentration of the ΔN25-FGF2 competitor. The competitor concentration promoting half-maximal inhibition ($IC_{50}$) of the signal was determined by fitting the experimental data with a non-linear regression model. Under the experimental conditions described here, the apparent $IC_{50}$ value corresponds to the dissociation constant ($K_D$) of the observed protein–protein interaction.

**Crosslinking experiments.** The crosslinking experiments shown in Fig. 3 were performed at 25 °C in 25 µl HK buffer (25 mM Hepes, pH 7.4; 150 mM KCl) at a final protein concentration of 10 µM. When FGF2 and α1-subCD3 were mixed, the final protein concentration was 20 µM, irrespective of the ratio between FGF2 and α1-subCD3 (1:1, 1:2, and 2:1). As chemical crosslinkers, BMOE

(bismaleidomethane) and DSG (disuccinimidyl glutarate) were dissolved in 20 mM DMSO and further diluted in HK buffer to 0.5 mM (BMOE) and 2 mM (DSG), respectively. Following preincubation of proteins for 30 min, reactions were started with the addition of BMOE or DSG yielding crosslinker/protein ratios of 1:1 (BMOE) and 4:1 (DSG). After 30 min, samples were quenched with 10 mM DTT (BMOE) or 20 mM Tris/HCl pH 7.5 (DSG), mixed with an equal volume of SDS sample buffer containing β-mercaptoethanol, and incubated for 10 min at 70 °C. Of each sample, 80% were analyzed on 1.5 mm NuPAGE 4–12% Bis-Tris gels (Invitrogen) and stained with Coomassie (InstantBlue, Expedeon). For a western analysis, 2% of each sample were separated on 1 mm NuPAGE 4–12% Bis-Tris gels. Blots were analyzed using affinity-purified anti-α1-CD1-3 rabbit antibodies[15] and monoclonal anti-FGF2 antibodies (clone bFM-1, Millipore).

**Structural analyses using NMR spectroscopy.** NMR spectra were recorded at 300 K on a Bruker Avance III 700 MHz spectrometer equipped with a 5 mm triple resonance cryo-probe. $^1$H-$^{15}$N-HSQC spectra were acquired of 77 µM $^{15}$N-labeled FGF2-C77/95S in the absence or presence of 70 µM or 138 µM α1-subCD3 in 25 mM HEPES (pH 7.4), 150 mM KCl, 10% $D_2O$ with 108 scans and 1024 data points in the $^1$H and 96 data points in the $^{15}$N dimension. Spectra were processed with Topspin3.2 (Bruker, Billerica, USA) and analyzed using CcpNMR Analysis 2.4.2 (ref. [45]). Peak assignments were transferred from published data [BMRB entry 4091 (ref. [42])] to nearest neighbors in recorded spectra, if possible, leading to an assignment of 63% of the non-Proline residues. Chemical shift differences were calculated using the formula $\Delta\delta(^1H,^{15}N) = (\delta(^1H)^2 + (0.15 \cdot \delta(^{15}N))^2)^{1/2}$. Signal-to-noise ratios were calculated from peak intensities and average noise levels. Overlapping peaks were omitted from the analysis.

**Multiple sequence alignment.** In order to identify residues that are uniquely present in FGF2 compared to signal-peptide-containing FGF family members, a multiple sequence alignment was performed using the EMBL-EBI MUSCLE 3.8 tool[46]. The protein sequence of isoform 3 of FGF2 (18 kDa form; UniprotKB ID P09038-2) was aligned with canonical sequences of FGF1 (P05230), FGF3 (P11487), FGF4 (P08620), FGF5 (P12034), FGF6 (P10767), FGF7 (P21781), FGF8 (P55075), FGF9 (P31371), FGF10 (O15520), FGF11 (Q92914), FGF12 (P61328), FGF13 (Q92913), FGF14 (Q92915), FGF16 (O43320), FGF17 (O60258), FGF18 (O76093), FGF19 (O95750), FGF20 (Q9NP95), FGF21 (Q9NSA1), FGF22 (Q9HCT0) and FGF23 (Q9GZV9). Supplementary Fig. 2 shows the part of the alignment that covers the residues 32–105 of FGF2.

**In silico docking studies.** The human sequence of the α1-subCD3 domain (T382–V599) was modeled based upon the existing crystal structure of the Na,K-ATPase of *Sus scrofa* [residues T380–V597, PDB ID: 3KDP[47]]. The two structures differ in only six amino acids, which is less than 3% of the total amino acid content. Point mutations using the CHARMM-GUI web server[48] were employed to model the human sequence of the α1-subCD3 domain. From the crystal structure of the Na,K-ATPase of Sus scrofa, the S391F, S473A, A497S, R500Q, I521L, and L578I substitutions were carried out to model the human structure of the α1-subCD3 domain.

Protein–protein docking studies were performed using the Rosetta 2018 package[49–52]. The α1-subCD3 domain was first considered as a spherical unit, and FGF2 was rotated around this sphere, positioning the surface area of FGF2 that contains K54 and K60 towards the α1-subCD3 surface. About 120,000 structures were generated with the Rosetta global docking protocol. About 94% of the structures were discarded based on low interface scores. All structures without contacts between K54 and K60 of FGF2 and the α1-subCD3 domain were discarded as well. Here, a contact was defined if the distance between any atoms of K54 or K60 in FGF2 and α1-subCD3 was less than 6 Å. The remaining structures were aligned to the α1-subCD3 domain of the full-length *Sus scrofa* crystal structure of Na,K-ATPase. All docked structures with overlaps of FGF2 and α1-subCD3 domains were also removed. This filtering procedure resulted in 62 candidates of docked structures for further analysis. For each of them, the Rosetta local docking protocol was employed to refine the global docking results. Therefore, for each of the 62 candidates, 500 structures were generated by randomly perturbing FGF2 by 3 Å translation and 8° rotation before the start of every individual simulation. These 31,000 structures were subjected to the same filtering procedure, which reduced the set to 33 structures. These were clustered based on the RMSD value for FGF2 using the Gromos algorithm[53] and a RMSD cut-off for two structures to be neighbors within 0.6 nm, which in turn resulted in four most populated clusters C1–C4. In each cluster, the most representative structure (as the centroid of the structures in a given cluster) was chosen as the basis to represent the wild-type (WT1–WT4), and the FGF2-α1-subCD3 interfaces of these structures were then tested and refined employing atomistic MD simulations. The first cluster (C1) was averaged from eleven structures, while the other three clusters (C2–C4) were averaged from four structures. These clusters had an averaged room-mean square deviation (RMSD) of 0.232, 0.146, 0.252, and 0.330 nm, respectively.

**Atomistic MD simulations.** The screening of the most probable FGF2/α1–subCD3 interaction interface was conducted through an in-depth study of the type of molecular interactions taking place during the simulation time and their residence period. The analysis of the MD simulations indicated the simulated WT1 cluster to represent the most stable FGF2/α1–subCD3 interface showing a broad range of

interactions together with three ion pairs, E525– K60, E498–R30, and K524–D27. Furthermore, K54 was found to be important as it stabilized the interaction between E498 and R30. As opposed to the WT1, the contact analysis for the WT2 system did not show stable contacts as revealed by MD simulations. In this case, only one pair of residues (K533–M150) was found with a probability of contacts higher than 50%. This suggests that the WT2 system is unlikely to be physiologically relevant. Similarly, the WT3 system appears to be unlikely to represent a relevant interface between FGF2 and α1-subCD3. While it was found to be stable based on MD simulations and its contact map showed contact frequencies of >50%, we did not find K54 and K60 to play a crucial role in stabilizing this potential protein–protein interaction interface. Despite the fact that the WT3 structure in the third cluster was characterized by a relatively large interaction interface, only one ion pair was found to stabilize this interface. In the case of the WT4 system, K54 and K60 were found to play a pivotal role as well as they made up about 25% of the whole contact interface between FGF2 and α1-subCD3. In this case, with E397– K54, E542–K60, E542-K153, and E548–K153, four electrostatic pairs were found to be involved in stabilizing the interaction between FGF2 and α1-subCD3. However, the network of molecular interactions in the WT4 system was not as extended as compared to observations in the WT1 system. In conclusion, the in silico results suggest the WT1 structure as the most probable protein–protein interaction surface between FGF2 and α1-subCD3 (Fig. 6).

The atomistic MD simulations were performed using the CHARMM36m[54] force field for lipids and proteins, the CHARMM TIP3P force field for water, and the standard CHARMM36 force field for ions. The GROMACS 2018.3 simulation package[55] was used in all simulations. For FGF2, we used its truncated structure [PDB ID: 1BFF[56]] from residue 26 to 154 in its monomeric form and the modeled version of human α1-subCD3. The N- and C-terminal groups were modeled as charged residues. The most representative structures of the four clusters were energy-minimized in vacuum using the steepest descent algorithm. The systems were first hydrated and neutralized by an appropriate number of counter-ions, followed by addition of 150 mM potassium chloride to mimic the experimental conditions. All systems were energy-minimized and an equilibration step was used to keep the temperature, pressure and the number of particles constant (NpT ensemble). During this step, proteins were restrained in all dimensions. For the production runs, all atoms in the region involved in the truncation part of α1-subCD3 to the rest of the Na,K-ATPase were restrained in all directions with a force constant of 1000 kJ/mol to avoid the unfolding of the α1-subCD3 domain. A second layer of restraints was applied to the alpha carbons of the residues in the FGF2/α1–subCD3 interaction interface and the thoroughly restrained part. No restraints were employed in the residues involved in the FGF2/α1–subCD3 interaction region. The Nose-Hoover thermostat[57] was used to maintain the temperature at 310 K with a time constant of 1.0 ps. The pressure of 1 atm was kept constant using the Parrinello–Rahman barostat[58] with a time constant set to 5.0 ps and isothermal compressibility to a value of $4.5 \times 10^{-5}$ bar$^{-1}$. The isotropic pressure-coupling scheme was used. For neighbor searching, we used the Verlet scheme with an update frequency of once every 20 steps. Electrostatic interactions were calculated using the Particle Mesh Ewald method[59]. Periodic boundary conditions were applied in all directions. The simulations were carried out using an integration time step of 2 fs until they reached 200 ns. All analyses were done for the last 100 ns of the 200 ns long simulation trajectories (unless stated otherwise) using standard GROMACS tools and in-house scripts. Finally, using the wild-type structures (WT1–WT4) as a basis, we constructed variant forms of FGF2 (K54E and K60E; systems M1–M4) and α1-subCD3 (D560N; system A1) by using the CHARMM-GUI web server. Each of these nine systems were simulated in the presence of one molecule of FGF2, one molecule of subCD3 and 42922 molecules of water for a total of 200 ns. A concentration of 150 mM of KCl salt was added to mimic experimental conditions. The systems were neutralized by an appropriate number of potassium atoms.

**Single-molecule TIRF microscopy**. Widefield fluorescence and TIRF images were acquired using an Olympus IX81 xCellence TIRF microscope equipped with an Olympus PLAPO ×100/1.45 Oil DIC objective lens and a Hamamatsu ImageEM Enhanced (C9100-13) camera. GFP fluorescence was excited with an Olympus 488 nm, 100 mW diode laser. Data were recorded and exported in Tagged Image File Format (TIFF) and analyzed via Fiji[60].

For the quantification of FGF2-GFP recruitment at the inner leaflet of the plasma membrane, cells were seeded in μ-Slide 8 Well Glass Bottom (ibidi) 24 h before live cell imaging experiments. The quantification of FGF2-GFP particles recruitment to the plasma membrane was achieved through the analysis of time-lapse TIRF movies. The frame of each cell was selected by widefield imaging. The number of FGF2-GFP particles were normalized to the cell surface area (μm²) and to the expression level of FGF2-GFP. The latter was quantified at the first frame of each time-lapse TIRF movie (for each analyzed cell) using ImageJ. The total number of FGF2-GFP particles per cell was quantified employing the Fiji plugin TrackMate[61]. Background fluorescence was subtracted in all the representative images and movies shown.

**Quantification of FGF2-GFP secretion from cells**. Quantification of FGF2-GFP on cell surfaces was done as described previously[20,23,43]. Stable CHO-K1 cell lines expressing various forms of FGF2-GFP in a doxycycline-dependent manner were cultured in α-MEM medium supplemented with 10% FCS, 2 mM glutamine, 100 U/ml penicillin, and 100 μg/ml streptomycin at 37 °C in the presence of 5% CO₂. Cells were seeded at $0.8 \times 10^5$ cells per ml in six-well plates (Corning Costar). Following 24 h of incubation, 1 μg/ml of doxycycline (Clontech) was added to induce FGF2-GFP expression. After a further 18 h of incubation, cells were washed twice with PBS supplemented with 1 mM MgCl₂ and 0.1 mM CaCl₂ and incubated with 1 mg/ml of a membrane-impermeable biotinylation reagent (EZ-Link Sulfo-NHS-SS-Biotin, Pierce; dissolved in 150 mM NaCl, 10 mM triethanolamine, pH 9.0, 2 mM CaCl₂) for 30 min at 4 °C. Following one washing step and 20 min of incubation with 100 mM glycine (dissolved in PBS supplemented with 1 mM MgCl₂ and 0.1 mM CaCl₂) at 4 °C, cells were washed twice with PBS and lysed at 37 °C with 1% Nonidet P-40 [in 50 mM Tris/ HCl, pH 7.5, 62.5 mM EDTA pH 8, 0.4% deoxycholate, protease inhibitor mixture (Roche Applied Science)]. Insoluble material was removed by centrifugation (10 min, 18,000 × g, 4 °C). Aliquots from the total cell lysate were taken and the remaining cell lysate was incubated with streptavidin beads (UltraLink immobilized streptavidin; Pierce) for 1 h at room temperature. Bound material was eluted with SDS sample buffer for 10 min at 95 °C. Total cell lysate and the biotinylated fraction were analyzed by western blotting using affinity-purified anti-GFP antibodies (1:500) and mono-clonal anti-GAPDH antibodies (Lifetech-Ambion; 1:20,000) as primary antibodies and fluorophore-labeled secondary antibodies (LI-COR anti-mouse and anti-rabbit; 1:10,000). Antigen signals were quantified using the Odyssey® CLx Imaging System (LI-COR Biosciences).

**DuoLink® proximity assays**. HeLa cells were cultured, processed and analyzed as described previously[15]. HeLa cells were grown on glass bottom culture dishes (MatTek 35 mm dishes with 10 mm microwell glass bottom). At a confluency of about 70%, cells were incubated for 2 h with indicated concentrations of ouabain or with DMSO as a mock control. Cells were then washed three times with PBS, fixed for 4 min with ice-cold acetone at −20 °C, and blocked with 1% BSA/PBS for 15 min at room temperature. Cells were incubated with the primary antibody solution (rabbit anti-FGF2 (1:500; ref. [20])) and mouse anti-α1 (1:200; Abcam ab7671) diluted in 1% BSA/PBS) for 1 h at room temperature. Cells were incubated with secondary antibodies conjugated to Duolink® In Situ PLA probes (Sigma-Aldrich) were diluted 1:5 in 1% BSA/PBS for 1 h at 37 °C. Ligation, amplification of DNA and its detection were conducted according to the manufacturer's manual using the DuoLink® detection reagent red (Sigma-Aldrich). Nuclei were stained with SYTOX green (1:75,000; Life Technologies) prior to imaging by confocal microscopy. DuoLink proximity signals obtained per cell were quantified using the Duolink® Image Tool software (Olink Bioscience).

The statistical analysis was based on five biological replicates each consisting of two technical replicates. Typically, 12–15 pictures with a total of about 150–300 cells were analyzed per technical replicate and condition. For each biological replicate, the average DuoLink proximity signals (number of red dots per cell as exemplified in Fig. 10a–c) of the mock condition was set to 100% and other experimental conditions were normalized accordingly. As a background control, experiments were conducted with either anti-FGF2 antibodies only or anti-α1 antibodies only and all data sets were corrected accordingly.

**Statistics and reproducibility**. This manuscript is accompanied by a file that contains all raw data from the studies contained in this work.

The statistical analyses were based on a one-way ANOVA test ($^{ns}p > 0.05$; $*p \le 0.05$; $**p \le 0.01$; $***p \le 0.001$). The exact $p$ values are indicated either directly in the figure or in the legend, unless $p \le 0.001$. The exact sample size is given in the legend of each figure. The mean ± standard deviation (SD) is displayed, unless otherwise stated.

In AlphaScreen competition experiments, the mean and the standard error of the mean (SEM) are given for each protein pair at the given concentration. The standard error is not displayed if it is smaller than the corresponding symbol. The IC₅₀ was determined from three biological replicates.

**Reporting summary**. Further information on research design is available in the Nature Research Reporting Summary linked to this article.

## Data availability
A supplementary file (Supplementary Data 1) with all raw data and statistical analyses contained in this work has been published along with the main publication and is available online. For all further queries, please contact the corresponding author (Walter Nickel; walter.nickel@bzh.uni-heidelberg.de)

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

## Acknowledgements

This work was funded by the Deutsche Forschungsgemeinschaft (DFG, German Research Foundation) as part of TRR 186 (Project Number 278001972), TRR 83, and Ni

423/7-1. This work was further supported by the DFG Cluster of Excellence CellNetworks at Heidelberg University, the Academy of Finland (Center of Excellence program), Helsinki Institute of Life Science (HiLIFE), and the Sigrid Juselius Foundation. We thank Monika Langlotz from the ZMBH Flow Cytometry Core Facility (Zentrum für Molekulare Biologie Heidelberg) and Holger Lorenz from the ZMBH Light Microscopy Core Facility (Zentrum für Molekulare Biologie Heidelberg) for their support in cell sorting and image acquisition/interpretation, respectively. C.F. and J.S. would like to acknowledge the assistance of the Core Facility BioSupraMol at Freie Universität Berlin, which is supported by the DFG. For computational resources, we wish to thank the CSC–IT Center for Science (Espoo, Finland). F.L. was supported by an HPC-Europa3 Transnational Access program (HPC175W35X).

## Author contributions

C.L., R.S., J.S., F.L., H.-M.M., S.W., E.D., and J.P.S. conducted the experiments and statistical analyses contained in this work. H.E., I.V., C.F., and W.N. designed the experiments, interpreted the data, and wrote the manuscript.

## Competing interests

The authors declare no competing interests.
