## [Peer Review File · Communications Biology]

Reviewers' comments:

Reviewer #1 (Remarks to the Author):

It has been known for many years that $\alpha 1$ NKA/ouabain regulate FGF2 secretion. However, the molecular mechanism of this regulation remains to be elusive. The authors took both biochemical and computational approaches and report the identification of two amino acid residues in FGF2 that appear to be directly involved in the interaction with $\alpha 1$ NKA. The data are convincing and of high quality. The experiments are well designed and performed. As suggested by the authors, it appears that $\alpha 1$ NKA works above PIPs in the regulation of FGF2 membrane targeting.

Although this clearly represents an incremental advancement in the fields of NKA, and potentially cancer biology, the impact of current work is limited. Specifically, the scope of work is quite narrow, most by GST pull-down assays. moreover, no structural elements have been identified in the $\alpha 1$ NKA that is important for the interaction. Most importantly, there was no attempt to provide molecular insight relating the identified interaction between FGF2 and $\alpha 1$ NKA to ouabain regulation of FGF2 secretion. Finally, no attempt was made to address the role of identified interaction in cell biology and animal pathophysiology.

In short, this could be a potentially interesting and important piece of work. however, much more needs to be done in order to significantly advance the fields.

Reviewer #2 (Remarks to the Author):

Legrand et al. studied the role of the $\alpha 1$ subunit of the Na,K-ATPase (ATP1A1) in unconventional secretion of the Fibroblast Growth Factor 2 (FGF2). They defined two lysine residues in the primary sequence of FGF2 for interaction with a subdomain of the cytoplasmic part of the α subunit. Substitution of either of these residues by glutamate or in combination impaired unconventional secretion of a GFP-tagged FGF2-reporter. The authors conclude that FGF2/ATP1A1 interaction precedes interaction of the growth factor with PI(4,5)P2 and facilitates its unconventional secretion across the membrane.

This is an interesting report that aims to elucidate molecular details and the sequence of events in unconventional secretion of FGF2 at the plasma membrane.

However, some issues should be addressed by the authors prior to publication:

1. The authors nicely demonstrated a role of the ATP1A1 subunit in FGF2-secretion in HeLa cells (Zacherl et al., 2015). Pharmacological experiments with COS-1 cells revealed similar conclusions (Florkiewicz et al., 1998). In this and in a previous study (Engling et al., 2002), they used CHO cells. However, control experiments with ATP1A1-depleted CHO K1 cells are missing to verify and strengthen their hypothesis.
2. Does the expression level of ATP1A1 vary between cell lines or tissues and is there a correlation in secretion efficiency of FGF2? This issue should at least be discussed.
3. The single particle TIRF microscopy experiments are elegant. Nevertheless, can the authors exclude recording of free diffusion of FGF2-GFP- or GFP-aggregates in the vicinity but not in association with the membrane? It appears as if even in the widefield images depicted in Fig. 9 punctate structures in the cytoplasm are visible.

Minor points:

4. Fig. 2A: indication of GST-tagged variant forms is confusing (GST- $\alpha 1$ CD3 appears twice)
5. Page 8, last paragraph: ... (Fig. 11C and 12(?)D). ...

Reviewer #3 (Remarks to the Author):

The tumor survival factor FGF2 is one of a growing number of proteins that are now known to be secreted from mammalian cells by 'non-conventional' direct transfer across the plasma membrane. Thanks to the work of the Nickel group the mechanistic basis of FGF2 transport is the best characterized of these processes.

The submitted manuscript fills in one of the remaining gaps in the understanding of FGF2 secretion by investigating why the plasma membrane enzyme Na,K-ATPase is required in this process. The authors define a binding site for FGF2 on the Na,K-ATPase and provide evidence that this allows the Na,K-ATPase to act as an early plasma membrane targeting factor during FGF2 secretion.

These are important results that are easily significant enough for publication in *Communications Biology*. As with other papers from the Nickel group, this highly multidisciplinary paper is extremely thorough and technically sound, and the manuscript is well presented. I strongly recommend publication after minor textural changes.

Specific points

The domain structure of the Na,K-ATPase that is being analysed in the paper would benefit from being more clearly presented at the start of the work. It would be good to see the domains shown on a 3D model of the Na,K-ATPase in Fig 1. Whilst I understand that the authors want to show structural detail in the later molecular modelling section (Fig 8), it would be helpful to the reader to have a better idea of how the Na,K-ATPase is being experimentally digested from the outset. There are many figures in the paper already so an additional panel is reasonable. If Fig 1 were to include a structure panel then it would be possible to annotate the positions of K54 and K60 on this so that the structural context can be understood at Fig 5 rather than having to wait until Fig 8. In the domain structure figure (Fig. 1A) it is not very clear that the red hashed area is the subdomain CD3. This could be set out in the Figure legend. The text on pg 4 describing the structural relationships of this subdomain is unclear ('This subdomain is contained in the third loop of the cytoplasmic domain of alpha-1 that is almost identical to the N-domain of the alpha-1 subunit') when the domain structure of the protein has not been set out/shown.

The pull-down data in Fig. 1 suggests some interactions of FGF2 with regions of Na,K-ATPase outside the CD3 domain. The raw AlphaScreen data in Fig 2A also suggest some interaction above the basal GST binding control, whereas the quantification in Fig 2B shows no difference from the control. Why do these data not agree? If one method is not more reliable than the other, then the statement on Pg 4 that there is the same binding to GST as to the CD3 domain deletion should be softened.

The first three paragraphs of the Discussion repeats much material from the introduction and should be trimmed considerably.

Point-by-point response to the reviewer's comments:

Reviewer 1:

While Reviewer 1 appreciated the high quality and conclusiveness of our study, she/he asked for additional experiments addressing molecular aspects of the role of the Na,K-ATPase in unconventional secretion of FGF2. From these suggestions, we chose two that are of direct relevance for this study and, therefore, strengthen the conclusions that can be drawn.

1.) Structural elements in the α 1 subunit of the Na/K ATPase engaged in the interaction interface with FGF2

Specific comment from Reviewer 1: ***“No structural elements have been identified in the α 1 NKA that is important for the interaction”***

As suggested by this reviewer, we have initiated studies on structural elements in the α 1 subunit that are important for its interaction with FGF2. These studies were based on molecular docking analyses and molecular dynamics simulations (new Fig. 9). They revealed a number of potential residues as candidates. As shown in the new Figure 10, using a quantitative protein-protein interaction assay based on AlphaScreen technology, a direct role of D560 in the cytoplasmic domain of α 1 could be confirmed to be part of this protein-protein interaction surface. These experiments provide the first insights into structural elements in the cytoplasmic domain of α 1 required for the interaction with FGF2. They further provide additional validation of the basic conclusions from our original manuscript and build the basis for future studies aiming at a comprehensive characterization of the molecular details of the interaction between α 1 and FGF2.

2.) The molecular mechanism by which Ouabain inhibits FGF2 secretion from cells

Specific comment from Reviewer 1: ***“There was no attempt to provide molecular insight relating the identified interaction between FGF2 and α 1 NKA to ouabain regulation of FGF2 secretion”***

Ouabain is a well-known inhibitor of the Na,K-ATPase. To obtain insight into the mechanism by which Ouabain inhibits unconventional secretion of FGF2, we conducted cell-based experiments probing proximity of α 1 and FGF2 at the plasma membrane in the absence and presence of Ouabain. As documented in the new Figure 14, based on a DuoLink proximity assay

for FGF2 and endogenous $\alpha 1$ we have established previously (Zacherl et al 2015, J Biol Chem), we quantified the proximity of FGF2 and $\alpha 1$ at the plasma membrane in the absence and presence of Ouabain. These experiments demonstrated that Ouabain inhibits FGF2/ $\alpha 1$ proximity events in cells in a dose-dependent manner. Our findings are consistent with previous experiments demonstrating that Ouabain does not inhibit interactions between a soluble form of the cytoplasmic domain of $\alpha 1$ and FGF2 (Zacherl et al 2015, J Biol Chem). This is because Ouabain binds to specific transmembrane spans of full-length $\alpha 1$. In the revised manuscript, these findings are discussed in terms of conformational changes exerted by Ouabain binding that are likely to lower the binding efficiency of FGF2 to $\alpha 1$. In conclusion, Ouabain inhibits the ability of FGF2 to bind to the cytoplasmic domain of $\alpha 1$ of the fully assembled Na,K-ATPase at the inner leaflet of the plasma membrane which, in turn, limits the ability of FGF2 to bind to PI(4,5)P₂, an event that occurs downstream of $\alpha 1$. This scenario is consistent with one of the major conclusions of our study, a requirement for FGF2 binding to the cytoplasmic domain of $\alpha 1$ for downstream events in this pathway that lead to PI(4,5)P₂-dependent FGF2 membrane translocation to the cell surface. With these findings, we also touch upon a third comment from Reviewer 1, requesting insights into the general role of the interaction of FGF2 and $\alpha 1$ in a cellular context as part of the unconventional secretory pathway of FGF2.

Reviewer 2:

1.) FGF2 secretion experiments in CHO cells depleted of the $\alpha 1$ subunit of the Na,K-ATPase

Specific comment from Reviewer 2: ***“The authors nicely demonstrated a role of the ATP1A1 subunit in FGF2-secretion in HeLa cells (Zacherl et al., 2015). Pharmacological experiments with COS-1 cells revealed similar conclusions (Florkiewicz et al., 1998). In this and in a previous study (Engling et al., 2002), they used CHO cells. However, control experiments with ATP1A1-depleted CHO K1 cells are missing to verify and strengthen their hypothesis.”***

It is correct that the original study that led to the identification of $\alpha 1$ as a molecular component of the unconventional secretory pathway of FGF2 was based on a genome-wide RNAi screen and subsequent validation in HeLa cells. This choice had been made because, from a technical point of view, HeLa cells are well suitable for these kinds of experiments as they allow for highly efficient down regulation of gene products using RNA interference. We indeed tried to conduct similar experiments in CHO cells, however, struggled with this approach because of relatively poor levels of down-regulation of $\alpha 1$. While we fully agree with Reviewer 2 that such experiments would have been useful, it wasn't possible to obtain conclusive data from CHO

cells due to these technical limitations. Nevertheless, with two independent data sets based on pharmacological inhibition (Ouabain; Engling et al 2002, J Cell Sci) and FGF2 variant forms with a defect in binding to $\alpha 1$ (the current study), we believe it is safe to conclude that the Na,K-ATPase is not only involved in unconventional secretion of FGF2 in COS-1 (Florkiewicz et al. 1998, J Biol Chem), HEK and CV-1 (Dahl et al 2000, Biochemistry) as well as HeLa cells (Zacherl et al 2015, J Biol Chem) but also in CHO cells. This view is also supported by the observation that all components so far identified to play a role in FGF2 secretion [PI(4,5)P₂, cell surface heparan sulfate proteoglycans, Tec kinase and the Na,K-ATPase] are ubiquitously expressed in a wide range of cell types. This matches the expression and secretion patterns of FGF2 from a similarly wide range of cells, in particular as part of developmental processes and a broad range of tumor cells.

2.) Correlation of Na,K-ATPase expression levels with FGF2 secretion efficiencies in different cell types

Specific comment from Reviewer 2: ***“Does the expression level of ATP1A1 vary between cell lines or tissues and is there a correlation in secretion efficiency of FGF2? This issue should at least be discussed.”***

There are no published studies that provide a systematic analysis correlating expression levels of the Na,K-ATPase with FGF2 secretion efficiencies. In our original studies in HeLa cells (Zacherl et al 2015, J Biol Chem), we used a number of different siRNAs targeting $\alpha 1$ that led to various degrees of down-regulation. From these studies, a clear correlation between $\alpha 1$ expression levels and FGF2 secretion efficiencies could not be deduced. Rather, it appeared that there is a certain threshold of $\alpha 1$ expression levels required for the efficient secretion of FGF2. These findings suggest that there is no linear relationship between these two parameters. Since the Na,K-ATPase is a relatively abundant component in basically all cell types, it probably cannot be expected to observe differences in FGF2 secretion efficiencies in response to different endogenous expression levels of $\alpha 1$.

3.) Association of FGF2-GFP with the inner plasma membrane leaflet as analyzed by TIRF microscopy

Specific comment from Reviewer 2: ***“The single particle TIRF microscopy experiments are elegant. Nevertheless, can the authors exclude recording of free diffusion of FGF2-GFP or GFP-aggregates in the vicinity but not in association with the membrane? It appears as if even in the widefield images depicted in Fig. 9 punctate structures in the cytoplasm are visible.”***

The single particle TIRF system we are using to quantify proximity of FGF2–GFP to the inner leaflet of the plasma membrane (Dimou et al 2019, J Cell Biol) uses GFP as a control. In this way, in the quantitative analysis of particle proximity to the inner leaflet, we correct for random diffusion of FGF2–GFP particles within the evanescent field analyzed by TIRF microscopy. Therefore, the data obtained represent a faithful quantification of FGF2–GFP particles that are retained in proximity of the inner plasma membrane leaflet. In the revised manuscript, we have improved the description of the TIRF experiments including quantification and the controls that were used.

All minor points raised by Reviewer 2 have been addressed in the revised manuscript.

Reviewer 3:

1.) Schematic representation of the domain structure of the $\alpha 1$ subunit of the Na,K-ATPase

Specific comment from Reviewer 3: ***“The domain structure of the Na,K-ATPase that is being analyzed in the paper would benefit from being more clearly presented at the start of the work. It would be good to see the domains shown on a 3D model of the Na,K-ATPase in Fig 1.”***

Based on the suggestion of Reviewer 3, the revised manuscript contains a remodeled version of Fig. 1. In addition to the schematic representation of the $\alpha 1$ constructs (panel A), it now contains 3D models of $\alpha 1$ (panel B) in which the various domains are highlighted in colors that correspond to the illustration of the constructs in panel A. The text describing Fig. 1 has been adjusted accordingly.

2.) Interpretations from pull-down and AlphaScreen data quantifying the interaction between FGF2 and $\alpha 1$

Specific comment from Reviewer 3: ***“The pull-down data in Fig. 1 suggests some interactions of FGF2 with regions of Na,K-ATPase outside the CD3 domain. The raw AlphaScreen data in Fig 2A also suggest some interaction above the basal GST binding control, whereas the quantification in Fig 2B shows no difference from the control. Why do these data not agree? If one method is not more reliable than the other, then the statement on Pg 4 that there is the same binding to GST as to the CD3 domain deletion should be softened.”***

The slight differences observed between the biochemical pull-down experiments and the AlphaScreen assays are most likely due to the fact that the former represents a rather semi-quantitative analysis. By contrast, AlphaScreen assays allow for a precise quantification of protein-protein interactions. As shown in the AlphaScreen assays in Fig. 8B, with regard to FGF2 binding efficiency, there is no significant difference between the GST negative control and GST- $\alpha 1$ -CD3 Δ sub. This is why we concluded that the $\alpha 1$ -subCD3 is the principal binding site for FGF2. Nevertheless, we fully agree with Reviewer 3 in that caution should be taken and also believe that elements beyond the sub-CD3 domain of $\alpha 1$ may have a small contribution regarding the interaction with FGF2. Therefore, as suggested by Reviewer 3, we have softened the corresponding statement in the revised manuscript.

3.) Repetitions in the Discussion

Comment from Reviewer 3: ***“The first three paragraphs of the Discussion repeat much material from the introduction and should be trimmed considerably.”***

With regard to a certain degree of repetitions between the introduction and the discussion, we have edited the manuscript carefully. However, we only made moderate cuts in the discussion as many readers directly jump to the Results and Discussion sections when reading a manuscript. Therefore, to make the manuscript accessible as much as possible, the discussion needs to recapitulate important background information about the work presented.

REVIEWERS' COMMENTS:

Reviewer #2 (Remarks to the Author):

All issues were adequately addressed.

Reviewer #3 (Remarks to the Author):

The revised version of the manuscript fully addresses the points that I raised in my initial review. I recommend publication.